

**Modern air, englacial and permafrost temperatures at high altitude on Mt.**
**Ortles, (3905 m a.s.l.) in the Eastern European Alps**
Luca Carturan[1], Fabrizio De Blasi[1,2], Roberto Dinale[3], Gianfranco Dragà[4], Paolo Gabrielli[5], Volkmar Mair[6],
Roberto Seppi[7], David Tonidandel[6], Thomas Zanoner[7,8], Tiziana Lazzarina Zendrini[1], Giancarlo Dalla
Fontana[1]
[1]Department of Land, Environment, Agriculture and Forestry, University of Padova, Viale dell'Università 16,
35020 Legnaro, Padova, Italy
[2]Consiglio Nazionale delle Ricerche - Istituto di Scienze Polari, c/o Ca' Foscari University of Venezia, Via
Torino 155, 30172 Mestre, Venezia, Italy
[3]Ufficio Idrografico, Provincia Autonoma di Bolzano, 39100 Bolzano, Italy
[4]Geo Monitoring Service s.r.l., Vicolo Santa Elisabetta 39, 39040 Varna, Bolzano, Italy
[5]Italian Glaciological Committee, c/o University of Turin, Via Valperga Caluso 35, 10125 Torino, Italy
[6]Ufficio Geologia e Prove materiali, Provincia Autonoma di Bolzano, 39053 Kardano, Bolzano, Italy
[7]Department of Earth and Environmental Sciences, University of Pavia, Via Ferrata 9, 27100 Pavia, Italy
[8]Department of Geosciences, University of Padova, Via Gradenigo 6, 35131 Padova, Italy
Correspondence to: luca.carturan@unipd.it

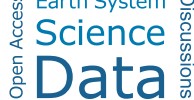

**Abstract**
The climatic response of mountain permafrost and glaciers located in high-elevation mountain areas has major
implications for the stability of mountain slopes and related geomorphological hazards, water storage and
supply, and preservation of paleoclimatic archives. Despite a good knowledge of physical processes that
govern the climatic response of mountain permafrost and glaciers, there is a lack of observational datasets from
summit areas. This represents a crucial gap in knowledge and a serious limit for model-based projections of
future behaviour of permafrost and glaciers.
A new observational dataset is available for the summit area of Mt. Ortles, which is the highest summit of
South Tyrol, Italy. This paper presents a series of air, englacial, soil surface and rock wall temperature collected
between 2010 and 2016. Details are provided regarding instrument type and characteristics, field methods,
data quality control and assessment. The obtained data series are available through an open data repository.
In the observation period the mean annual air temperature at 3830 m a.s.l. was between -7.8 and -8.6°C. The
most shallow layers of snow and firn (down to a depth of about 10 m) froze during winter. However melt water
percolation restored isothermal conditions during the ablation season and the entire firn layer was found at the
melting pressure point. Glacier ice is cold, however only from about 30 m depth. Englacial temperature
decreases with depth reaching a minimum of almost -3°C close to the bedrock, at 75 m depth. A small glacier
located on a rocky ridge of Mt. Ortles at 3470 m a.s.l., without firn cover, was also found in cold conditions
from the surface down to a depth of 9.5 m. The mean annual ground surface temperature was negative for all
but one monitored sites, indicating cold ground conditions and the existence of permafrost in nearly all debris-
mantled slopes of the summit. Similarly, the mean annual rock wall temperature was negative at most
monitored sites, except the lowest one at 3030 m a.s.l. This suggests that the rock faces of the summit are
affected by permafrost at all exposures.








## 1. Introduction

High-elevation mountain areas are complex systems influenced by physical processes occurring in the atmosphere, cryosphere and lithosphere. These processes closely interact and govern the energy and mass balance and climatic response of mountain permafrost and glaciers located at high elevation. Their response to climatic changes has important consequences for i) the stability of mountain slopes and related geomorphological hazards (Huggel et al., 2015; Knight and Harrison, 2023), ii) the thermal regime, water storage and stability of mountain glaciers (Deline et al., 2015), iii) the hydrological balance and water supply from glacierized catchments (Irvine-Fynn and Hubbard, 2017), and iv) the formation and preservation of paleoclimatic archives, such as glacier geochemical records (Gabrielli et al., 2010).

The ongoing atmospheric warming is leading to a deep transformation of these high-elevation systems, which react sensitively to climatic changes. Indeed, the thermal state of the cryosphere is strongly influenced by variations in air temperature, which regulates its energy and mass balance and dynamic behaviour (Harris et al., 2009; Cicoira et al., 2019; Deline et al., 2015). Projections of future global climate indicate further warming in absence of mitigation policies such as the reduction of greenhouse gas emission (IPCC, 2022). For this reason, the current impacts on high-elevation mountain areas are expected to continue and possibly accelerate.

Direct observations of the thermal state and response of high-elevation mountain areas are of great importance. Even though the physical processes that govern the energy and mass balance and climatic response of mountain permafrost and glaciers are known, model-based projections of their future behaviour are subject to large uncertainty. This is because the observational datasets required for model calibration and validation are particularly scarce for these summit areas, where model inputs and results are often poorly constrained and extrapolated, in absence of direct observations (Charbonneau et al., 1981; Machguth et al., 2008; Carturan et al., 2012).

Thermal observations in high-elevation mountain areas are also of great value for i) improving knowledge on the air temperature variability (e.g. the so-called elevation-dependent warming, Pepin et al., 2015 and 2022), or the glacier cooling effect (Braithwaite et al., 2002), ii) better understanding the relationship between climatic proxies and meteorological variables (e.g. ice cores, Bohleber et al., 2013), iii) evaluating/improving models (e.g. permafrost distribution models, Boekli et al., 2012), iv) biological and biogeochemical studies (e.g. Rathore et al., 2018), and v) setting baseline conditions for future studies and trend analyses.



In this paper we present a novel six-year dataset of air, rock, soil surface and englacial temperature collected
between 2010 and 2016 on the summit of Mt. Ortles (46.508° N, 10.541° E, 3905 m a.s.l.), in the eastern Italian
Alps. These observations were carried out in the framework of the Ortles Project (ortles.org; Gabrielli et al.,
2016). This is an international research project, coordinated by the Byrd Polar and Climate Research Center,
The Ohio State University (USA) and the Hydrographic Office of the Autonomous Province of Bolzano, with
the aim of extracting ice cores from the Alto dell'Ortles Glacier (Oberer Ortlerferner) to be used for
paleoclimatic and paleoenvironmental investigations.
Here we provide a full description of the experimental site, data-collection methods and equipment, raw data
processing and final datasets.

**2. Site description**
Mount Ortles (46.508° N, 10.541° E, 3905 m a.s.l.) is located in the Ortles-Cevedale Mountain Group, which
is the largest glacierized area in the Italian Alps (Carturan et al., 2013). It is the highest peak of South Tyrol,
in the eastern European Alps (Fig. 1). From a lithological point of view, the summit of Mt. Ortles is mainly
composed by dolomites, alternated with dark-stratified limestones and paraconglomeratic limestone levels and
breccias. Local outcrops of phyllites rich in quartz and orthogneiss can be found at the base of the mountain
(Montrasio et al., 2012).
The northern part of the Ortles-Cevedale mountain range is characterised by a continental climate, with scarce
annual precipitation (500 mm in the lower valley), which falls mostly in summer. Towards the south, there is
an increasing Mediterranean influence and the annual precipitation maxima are in spring and autumn, with
cumulative amounts of 900 mm in the lower valleys. In the glaciated areas in the middle of the mountain range,
at 3000-3200 m a.s.l., the mean annual precipitation has been estimated between 1400 and 1500 mm (Carturan,
2010; Carturan et al., 2012). The mean annual isotherm of 0°C is located at about 2500 m a.s.l. At the elevation
of the glaciers (above 3000 m) the snow cover shows a typical annual cycle, with the accumulation season
occurring between October and May, and the ablation season between June and September. On the glaciers,
however, snowfalls are frequent during summer, especially above 3300-3500 m.
Glaciers, glacierets and snowfields cover the Mt. Ortles flanks. Here we describe the two ice bodies
investigated in the Ortles Project. The summit area is almost entirely covered by the Alto dell'Ortles Glacier



(Oberer Ortlerferner), which is the highest glacier of South Tyrol, ranging in altitude between 3018 and 3905
m a.s.l. and covering an area of 1.04 km$^2$ (year 2008). The observed glacier thickness is about 75 m (Gabrielli
et al., 2012) and the vertical ice profile encompasses the last ~7 kyr (Gabrielli et al., 2016). This glacier is
polythermal, with temperate firn and cold ice underneath (Gabrielli et al., 2012). From geomorphological
evidence (trimlines and moraines) it is possible to estimate a maximum Little Ice Age (14[th] - 19[th] centuries)
area of 2.09 km$^2$ for this glacier, and a 50% area loss since then. Between 1984 and 2005 the (geodetic) mass
balance of the glacier was closer to equilibrium (-0.18 m w.e. y$^{-1}$) when compared to the majority of glaciers
in the Ortles-Cevedale Group in the same period (mean balance rate of -0.69 m w.e. y$^{-1}$, Carturan et al., 2013).
A small glacier, named Hintergrat, covers part of the eastern rocky ridge of Mt. Ortles. The area of this glacier
is 0.09 km$^2$ and its elevation ranges between 3340 and 3580 m a.s.l. This glacier is mostly in cold thermal
conditions and its front hangs over the Fine del Mondo Glacier (Ende der Welt Ferner) underneath.
Mountain permafrost is widespread on Mt. Ortles, according to the permafrost distribution modelled by
Boeckli et al. (2012), that indicates 'permafrost in nearly all conditions' above 2600 m for areas with northern
exposure, and above 2900 m for areas with southern exposure.
Before the Ortles Project, no specific investigation existed on the air, englacial and permafrost temperatures
of this mountain.
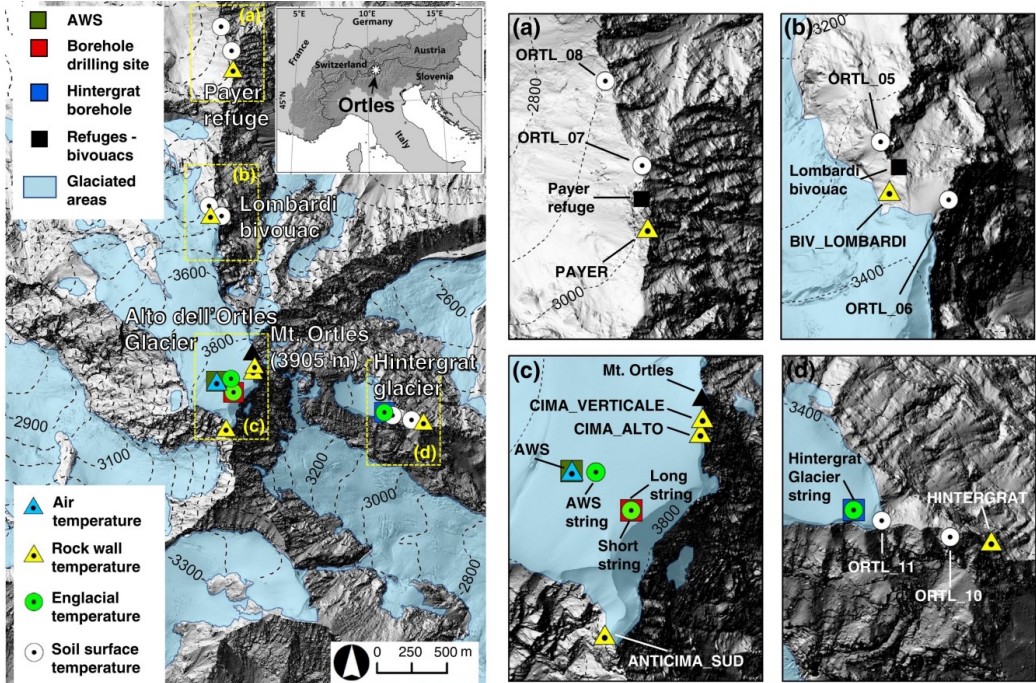


Figure 1. Geographic location of Mt. Ortles and of sites instrumented for air, rock, englacial and soil surface
temperature measurements. Close ups of the a) Payer Refuge, b) Lombardi Bivouac, c) Mt. Ortles summit
area, and d) Hintergrat ridge are reported in the panes on the right. The background hillshaded DEM is from
http://geocatalogo.retecivica.bz.it/.

## 3. Data description


The temperature datasets presented in this work were obtained by installing stand-alone dataloggers connected
to one or several temperature sensors. Due to the remoteness of the study site, dataloggers were powered by
lithium or lead-acid batteries, which in the case of the Automatic Weather Station (AWS, Section 3.1) were
recharged daily by solar panels. Periodic field visits, mostly performed from June to September, were used for
instrumentation maintenance and data download using a laptop. No real-time transmission of data was setup.
The dataset is characterised by a good time coverage and a few gaps (Fig. 2), indicating the suitability of the
selection of the equipment and field procedures for installation and maintenance. The most significant temporal
gaps affect soil surface temperature datasets and were caused by the impossibility of accessing dataloggers in
late summer of 2011. Other minor gaps were caused by temporary malfunctions or by damaged equipment,



e.g., the rupture of the fan-aspirated radiation shield at the AWS between February and August 2013, which
forced us to treat this period as a data gap. We did not undertake gap-filling, in order to keep the data recorded
in the field as unchanged as possible.
Details of measuring equipment and installations are provided in the following sections. Further details about
the instruments are provided in Table 1 and a topographic description of instrumented sites is reported in Table
E1, in the Appendix. Figures showing examples of data series for each variable are provided in the following
subsections.

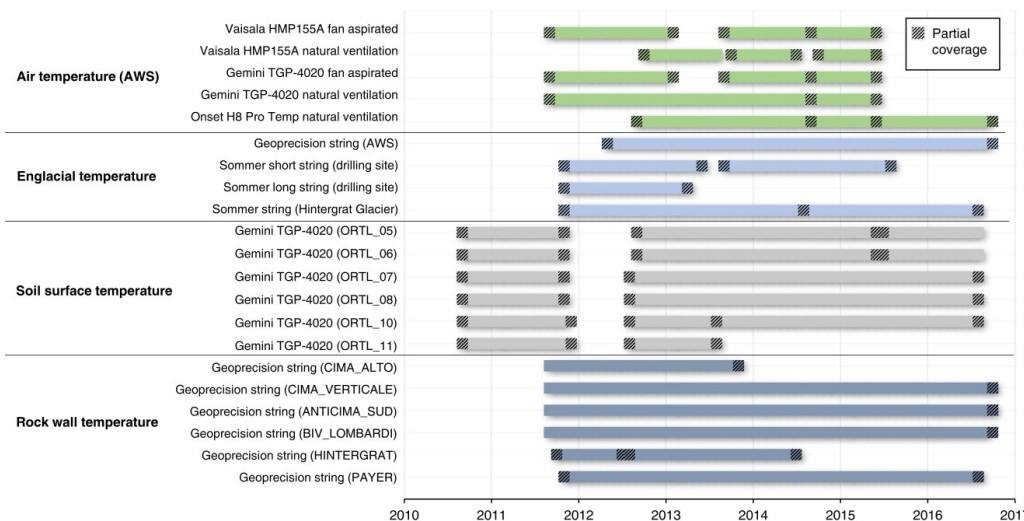


Figure 2. Monthly coverage of temperature measurements available from 2010 to 2016 on Mt. Ortles. Partial
coverage indicates the occurrence of data gaps for specific months.


Table 1. Sensor characteristics, setup and period of operation for air, englacial, soil surface and rock wall
temperature measurements on Mt. Ortles.

| Measured variable | Sensor | Radiation shield | Period of operation | | Initial height (m) | Unit | Interval | Integration method and interval | Accuracy |
|---|---|---|---|---|---|---|---|---|---|
| | | | from | to | | | | | |
| **Air temperature data (AWS)** | | | | | | | | | |



| | | | | | | | | | |
|---|---|---|---|---|---|---|---|---|---|
| Air Temperature | Vaisala HMP155A | R. M. Young 43502 fan-aspirated radiation shield | Sep 2011 | Jun 2015 | +4 | °C | 15 min | avg 1 h | ±(0.226 - 0.0028·T )°C from -80 to 20°C, ±(0.055 + 0.0057·T)°C from 20 to 60 °C |
| Air Temperature | Vaisala HMP155A | Campbell Scientific MET 21 radiation shield with natural ventilation | Sep 2012 | Jun 2015 | +4 | °C | 15 min | avg 1 h | ±(0.226 - 0.0028·T )°C from -80 to 20°C, ±(0.055 + 0.0057·T)°C from 20 to 60 °C |
| Air Temperature (Backup and comparison) | Gemini TGP-4020 | R. M. Young 43502 fan-aspirated radiation shield | Sep 2011 | Jun 2015 | +4 | °C | 1 h | avg 1 h | ±(0.2 - 0.005·T )°C from -40 to 0°C, ±0.2 °C from 0°C to 40°C |
| Air Temperature (Backup and comparison) | Gemini TGP-4020 | R.M. Young 41303-5 radiation shield with natural ventilation | Sep 2011 | Jun 2015 | +4 | °C | 1 h | avg 1 h | ±(0.2 - 0.005·T )°C from -40 to 0°C, ±0.2 °C from 0°C to 40°C |
| Air Temperature (Backup and comparison) | Onset Hobo H8 Pro Temp | Davis 7714 radiation shield with natural ventilation | Sep 2012 | Sep 2016 | +4 | °C | 1 h | avg 1 h | ±(0.63 - 0.022·T )°C from -40 to 0°C, ±0.63 °C from 0°C to 40°C |
| **Englacial temperature data** | | | | | | | | | |
| Snow and firn temperature at the AWS site | Geoprecision thermistor string (15 sensors) | | Jun 2012 | Sep 2016 | -0.6 / -1.6 / -2.6 / -3.6 / -4.6 / -5.6 / -6.6 / -7.6 / -8.6 / -9.6 / -10.6 / -11.6 / -12.6 / -13.6 / -14.6 | °C | 2 h | instant 2 h | ±0.5°C from -10°C to +85°C |
| Englacial temperature at the borehole drilling site (short string) | Sommer thermistor string (11 sensors) | | Nov 2011 | Aug 2015 | 0 / -1 / -2 / -3 / -4 / -6 / -8 / -10/ -15 / -20 / -25 | °C | 1 h | instant 1h | ±0.1°C from 0°C to 70°C |
| Englacial temperature at the borehole drilling site (long string) | Sommer thermistor string (4 sensors) | | Nov 2011 | Apr 2013 | -15 / -35 / -55 / -75 | °C | 1 h | instant 1h | ±0.1°C from 0°C to 70°C |
| Englacial temperature at the Hintergrat Glacier | Sommer thermistor string (5 sensors) | | Nov 2011 | Aug 2016 | -1.5 / -3.5 / -5.5 / -7.5 / -9.5 | °C | 1 h | instant 1h | ±0.1°C from 0°C to 70°C |
| **Soil surface temperature data** | | | | | | | | | |
| Soil surface temperature at Lombardi bivouac - ORTL_05 | Gemini TGP-4020 | | Sep 2010 | Sep 2016 | -0.15 | °C | 1 h | avg 1h | ±(0.2 - 0.005·T )°C from -40 to 0°C, ±0.2 °C from 0°C to 40°C |
| Soil surface temperature at Lombardi bivouac - ORTL_06 | Gemini TGP-4020 | | Sep 2010 | Sep 2016 | -0.12 | °C | 1 h | avg 1h | ±(0.2 - 0.005·T )°C from -40 to 0°C, ±0.2 °C from 0°C to 40°C |
| Soil surface temperature at Payer refuge - ORTL_07 | Gemini TGP-4020 | | Sep 2010 | Aug 2016 | -0.05 | °C | 1 h | avg 1h | ±(0.2 - 0.005·T )°C from -40 to 0°C, ±0.2 °C from 0°C to 40°C |
| Soil surface temperature at Payer refuge - ORTL_08 | Gemini TGP-4020 | | Sep 2010 | Aug 2016 | -0.05 | °C | 1 h | avg 1h | ±(0.2 - 0.005·T )°C from -40 to 0°C, ±0.2 °C from 0°C to 40°C |
| Soil surface temperature at Hintergrat ridge - ORTL_10 | Gemini TGP-4020 | | Sep 2010 | Aug 2016 | -0.05 | °C | 1 h | avg 1h | ±(0.2 - 0.005·T )°C from -40 to 0°C, ±0.2 °C from 0°C to 40°C |
| Soil surface temperature at Hintergrat ridge - ORTL_11 | Gemini TGP-4020 | | Sep 2010 | Aug 2013 | -0.05 | °C | 1 h | avg 1h | ±(0.2 - 0.005·T )°C from -40 to 0°C, ±0.2 °C from 0°C to 40°C |



**Rock wall temperature data**

| | | | | | | | | |
|---|---|---|---|---|---|---|---|---|
| Rock wall temperature at Mt. Ortles summit - CIMA_ALTO | Geoprecision thermistor string (3 sensors) | Sep 2011 | Nov 2013 | -0.10 / -0.30 / -0.55 | °C | 1 h | instant 1h | ±0.5 °C from -30 to -5 °C, ±0.1 °C from -5 to +40 °C |
| Rock wall temperature Mt. Ortles summit - CIMA_VERTICALE | Geoprecision thermistor string (3 sensors) | Sep 2011 | Sep 2016 | -0.10 / -0.30 / -0.55 | °C | 1 h | instant 1h | ±0.5 °C from -30 to -5 °C, ±0.1 °C from -5 to +40 °C |
| Rock wall temperature at Vorgipfel - ANTICIMA_SUD | Geoprecision thermistor string (3 sensors) | Sep 2011 | Sep 2016 | -0.10 / -0.30 / -0.55 | °C | 1 h | instant 1h | ±0.5 °C from -30 to -5 °C, ±0.1 °C from -5 to +40 °C |
| Rock wall temperature at Lombardi bivouac - BIV_LOMBARDI | Geoprecision thermistor string (3 sensors) | Sep 2011 | Sep 2016 | -0.10 / -0.30 / -0.55 | °C | 1 h | instant 1h | ±0.5 °C from -30 to -5 °C, ±0.1 °C from -5 to +40 °C |
| Rock wall temperature at Hintergrat - HINTERGRAT | Geoprecision thermistor string (3 sensors) | Oct 2011 | Aug 2014 | -0.10 / -0.30 / -0.55 | °C | 1 h | instant 1h | ±0.5 °C from -30 to -5 °C, ±0.1 °C from -5 to +40 °C |
| Rock wall temperature at Payer refuge - PAYER | Geoprecision thermistor string (3 sensors) | Nov 2011 | Aug 2016 | -0.10 / -0.30 / -0.55 | °C | 1 h | instant 1h | ±0.5 °C from -30 to -5 °C, ±0.1 °C from -5 to +40 °C |




### *3.1 Air temperature data*


On 30 September 2011 an Automatic Weather Station (AWS) was installed on the upper accumulation area of
Alto dell'Ortles Glacier, at an elevation of 3830 m a.s.l. The AWS was equipped with a Campbell Scientific
CR-1000 datalogger, solar panels and sensors for air temperature and relative humidity (Vaisala HMP155A),
wind speed and direction (R. M. Young 05103), incoming and outgoing shortwave and longwave radiation
(Delta Ohm LP Pyra 05 and LP PIRG 01), and snow depth (Campbell Scientific SR50A). The equipment was
supported by an aluminum tower (composed of 2-m modules), anchored in the firn at 2 m depth and supported
by wooden boards. After the installation, the tower extended 4 m from the surface. The sensors and solar panels
were fixed on top of the tower, whereas the datalogger/battery housing box was fixed at the bottom (Figs. F1
and F2).
The Vaisala HMP155A sensor was installed inside a R. M. Young 43502 fan-aspirated radiation shield. Two
standalone Gemini TGP-4020 dataloggers equipped with PB-5003-1 thermistor probes were also installed for
comparison and backup purposes, one in the same fan-aspirated radiation shield of the Vaisala HMP155A
sensor and one inside a 6-plate R.M. Young 41303-5 radiation shield with natural ventilation. In September

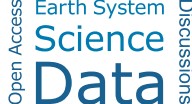

2012, we installed an additional HMP155A temperature sensor inside a 15-plate Campbell Scientific MET 21
radiation shield with natural ventilation, and an Onset Hobo H8 Pro Temp datalogger housed in an 8-plate
Davis 7714 radiation shield with natural ventilation (Figs. F3 and F4). Sensor specifications are reported in
Table 1. The Vaisala HMP155A data were recorded as 15-minute mean values, whereas the Gemini TGP-4020
dataloggers recorded hourly minimum and maximum temperature, and the Hobo H8 Pro Temp datalogger
recorded hourly instantaneous temperature. All the temperature records have been converted into hourly
averages, averaging 15-minute means for the Vaisala HMP155A sensors, minimum and maximum hourly
temperature for the Gemini TGP-4020 dataloggers, and instantaneous temperature at the beginning and at the
end of each hour for the Hobo H8 Pro Temp datalogger, assuming a linear variation of temperature during
each hour.
All the air temperature sensors were installed at the same level (±20 cm). The height of the air temperature
sensors above the glacier surface changed with the snow accumulation over time. To prevent burial by snow
accumulation, the tower was elongated annually by adding a 2-m module. Figure 3 shows the height of the
sensors above the glacier surface, as reconstructed from the snow depth data and maintenance logs (Table D1).

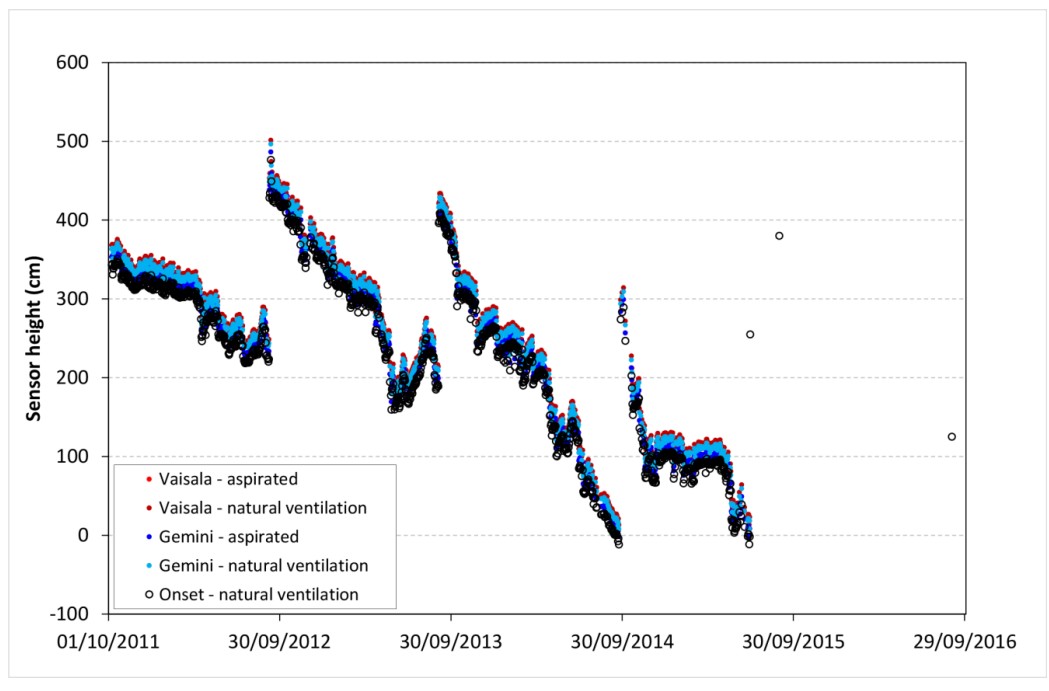


Figure 3. Air temperature sensors height above the snow surface.


Despite the remote and harsh environment, the AWS worked properly without major interruptions. In June
2015 it was removed, but the Hobo H8 Pro Temp logger was left on site and recorded additional 15 months of
data. The main issue linked to the specific environment of installation was ice and snow accretion combined
with strong winds, which damaged the fan-aspirated radiations shield in February 2013. The obtained data are
shown in Fig. 4.


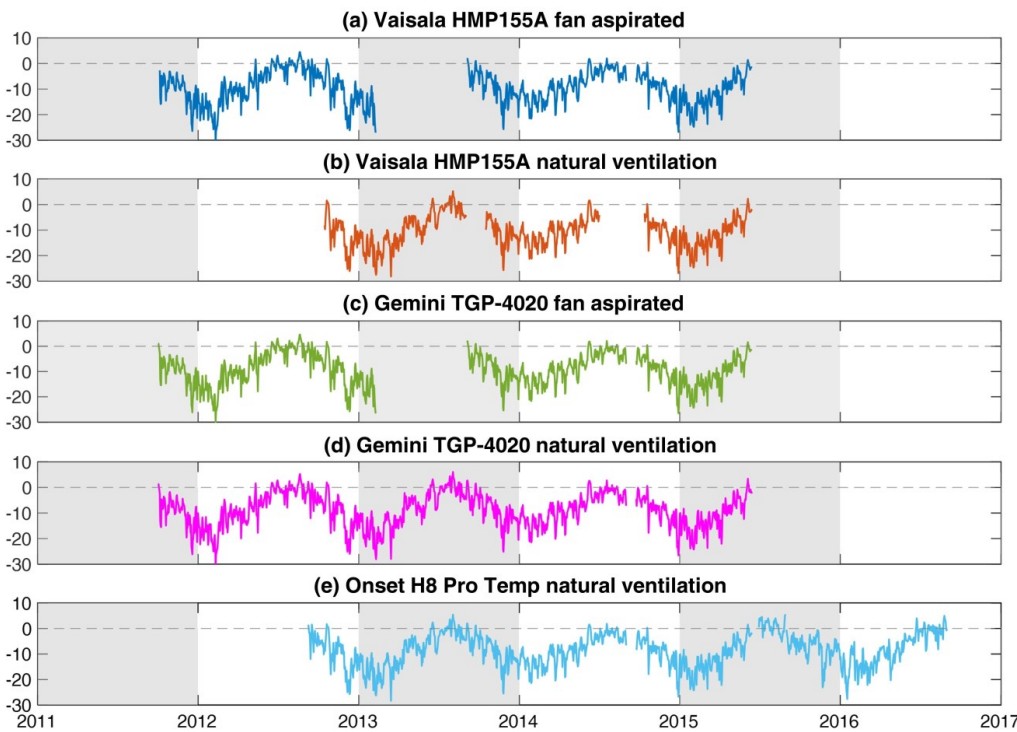


Figure 4. Daily mean air temperature series measured by different sensors installed at the Automatic Weather
Station on Mt. Ortles.



### *3.2 Englacial temperature data*
Englacial temperature measurements were collected at three different sites on Mt. Ortles: i) at the AWS site
(3830 m), ii) at the borehole drilling site (3859 m), and iii) at the Hintergrat glacier (3476 m). The obtained
englacial temperature data are shown in Fig. 5.

#### 3.2.1 Snow and firn temperature data at the AWS site
On the 18th of June 2012 a 20-meter thermistor string manufactured by Geoprecision GmbH (Germany) was
installed 10 m east of the AWS (Fig. F1). The thermistor string was composed of a Dallas M-Log5W
datalogger, powered by a 3.6 V lithium battery, and connected to 15 digital Dallas temperature sensors spaced
one meter from each other. The string was lowered into a 14.6 m hole drilled using a steam ice drill. The initial
depth of temperature sensors ranged between 0.6 and 14.6 m, and increased afterwards up to about 6 m due to
the accumulation of snow. The logger was housed inside a plastic box on the glacier surface, subsequently
buried in the snow. Instantaneous temperature data were recorded with a 2-hour frequency.
The data were retrieved by means of a laptop using a USB dongle connected wireless (radio transmission) to
the logger, below the glacier surface. We were able to retrieve temperature data with the logger buried below
a maximum of ~6 m of snow and firn. The thermistor string worked properly without interruptions and without
requiring maintenance or battery replacement. Sensor specifications are reported in Table 1.

#### 3.2.2 Englacial temperature at the borehole drilling site
The site where the Ortles Project deep ice cores were extracted is a small col (3859 m; 10°32'34", 46°30'25")
between the summit of Mt. Ortles and the Anticima Sud/Vorgipfel (3845 m, Figs. 1 and G1). The ice is 75 m
thick at this site as indicated by geophysical sensing prospecting and confirmed by ice core drilling operations
(Gabrielli et al., 2016). Two thermistor strings were installed in borehole number 3 on the 5th of October 2011,
immediately after the completion of the drilling operations (Fig. G3a). The strings were composed of an MDL
8/3 datalogger, manufactured by Sommer GmbH & Co KG (Austria). The logger was connected to 44031
thermistors, manufactured by ThermX (USA).

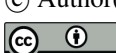



A first (short) thermistor string was 35 m long, and was equipped with temperature sensors at 0, 1, 2, 3, 4, 6,
8, 10, 15, 20, 25 m (initial depth). The other (long) string was 100 m, with temperature sensors at 15, 35, 55
and 75 m (initial depth). Burial depth of sensor increased over time due to net snow and firn accumulation.
Dataloggers and exceeding portions of strings were housed inside a metal box and arranged on a winding
system (Fig. G3), making it possible to extend the thermistor strings and to arise the box at the glacier surface
periodically. Field maintenance was also required to replace batteries and download the stored data.
Instantaneous temperature data were recorded with a 1-hour frequency.
The short string worked properly until removal in summer 2015, with the exception of a two-month gap in
summer 2013. The long string stopped working in April 2013, possibly due to ice dynamics and deformation
of the borehole at a depth below 25-30 m. Sensor specifications are reported in Table 1.

3.2.3 Englacial temperature at the Hintergrat Glacier
On the 14th of October 2011 a thermistor string was installed at 3476 m a.s.l. on top of the Hintergrat Glacier
(Fig. G2). The string was manufactured by Sommer GmbH & Co KG (Austria), with the same components as
those installed at the ice core drilling site (Section 3.2.2, Fig. G4). It was lowered into a hole drilled using a
steam ice drill down to a depth of 9.6 m. We did not reach the glacier bottom at this site. The temperature
sensors were placed at 1.5, 3.5, 5.5, 7.5 and 9.5 m below the glacier surface. This lower site is subject to net
ablation, therefore in this case the initial depth decreased through time and the sensor at 1.5 m depth came to
the surface in summer 2013. After 2013, this sensor's data were consequently discarded.





Figure 5. Englacial temperature measured at various sites and various depths on the Alto dell'Ortles Glacier.

### 3.3 Soil surface temperature data

The thermal regime of the soil surface at six deglaciated sites on Mt. Ortles was monitored using standalone

temperature dataloggers over the period between September 2010 and September 2016. We used Gemini TGP-



4020 dataloggers, powered by 3.6 V lithium batteries, and equipped with PB-5001 probes, which were placed
5-15 cm below the soil surface (Figs. H1 and H2). Mean temperature data were recorded at hourly intervals.
Periodic maintenance was required to download the data and replace exhausted batteries.
The monitored sites range in elevation between 2899 and 3466 m a.s.l. The dataloggers were placed in pairs
at three main locations (Figs. 1 and H3): refuge Payer (ORTL_07 and ORTL_08), bivouac Lombardi
(ORTL_05 and ORTL_06) and Hintergrat ridge (ORTL_10 and ORTL_11).
The data series extend from late summer 2010 to late summer 2016, with a gap between autumn 2011 and late
summer 2012 and for ORTL_05 and ORTL_06 also between July and August 2015, due to the impossibility
of accessing the dataloggers for maintenance. ORTL_11 was buried under snow and firn after 2013 and has
never been recovered. The obtained soil surface temperature data are displayed in Fig. 6.

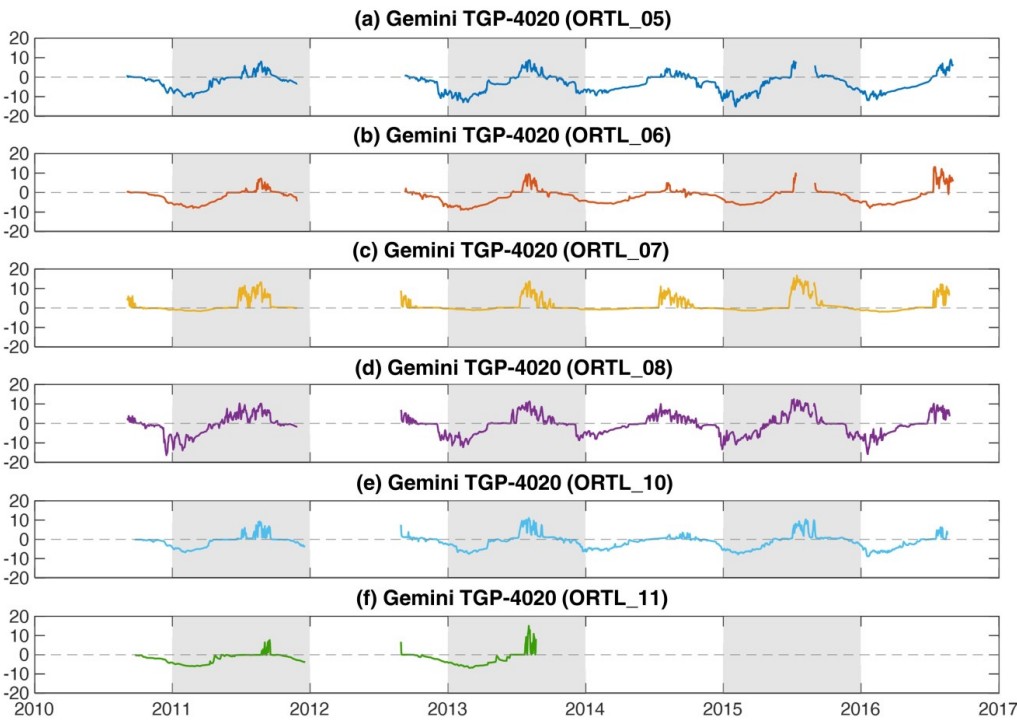

Figure 6. Soil surface temperature data measured at various locations on Mt. Ortles.





### *3.4 Rock wall temperature data*

The sub-surface temperature of rock walls located at six sites on Mt. Ortles was monitored starting in late
summer and autumn 2011 (Fig. 2). Very steep/almost vertical rock walls with different exposures and
elevations were selected for monitoring (Figs. 1 and I3). Two sites were established next to the Mt. Ortles
summit (3900 and 3880 m, facing East), one at the Vorgipfel (3844 m, facing South), one at the Hintergrat
(3370 m, facing North-East), one at the bivouac Lombardi (3351, facing West) and one at the refuge Payer
(3030 m, facing East).
Rock temperature data were acquired using Geoprecision Dallas M-Log5W dataloggers, powered by a 3.6 V
lithium battery, and connected to three digital Dallas temperature sensors installed at 0.1, 0.3 and 0.55 m depth,
into holes drilled with a hammer drill (Fig. I1). Instantaneous temperature data were stored at hourly intervals
and downloaded with a remote connection using a wireless USB dongle and a laptop (Fig. I2).
The datalogger placed at the Hintergrat was damaged by hikers in late July 2012. After the damage was
repaired in late August 2012 it was operational until August 2014 when it was again found badly damaged and
therefore it was removed.
One of the loggers installed at the Mt. Ortles summit ("CIMA_ALTO") stopped working in November 2013
due to battery failure, the other dataloggers worked properly until the end of the monitoring period, in late
summer 2016. Sensor specifications are reported in Table 1. The obtained rock wall temperature data are
displayed in Fig. 7.

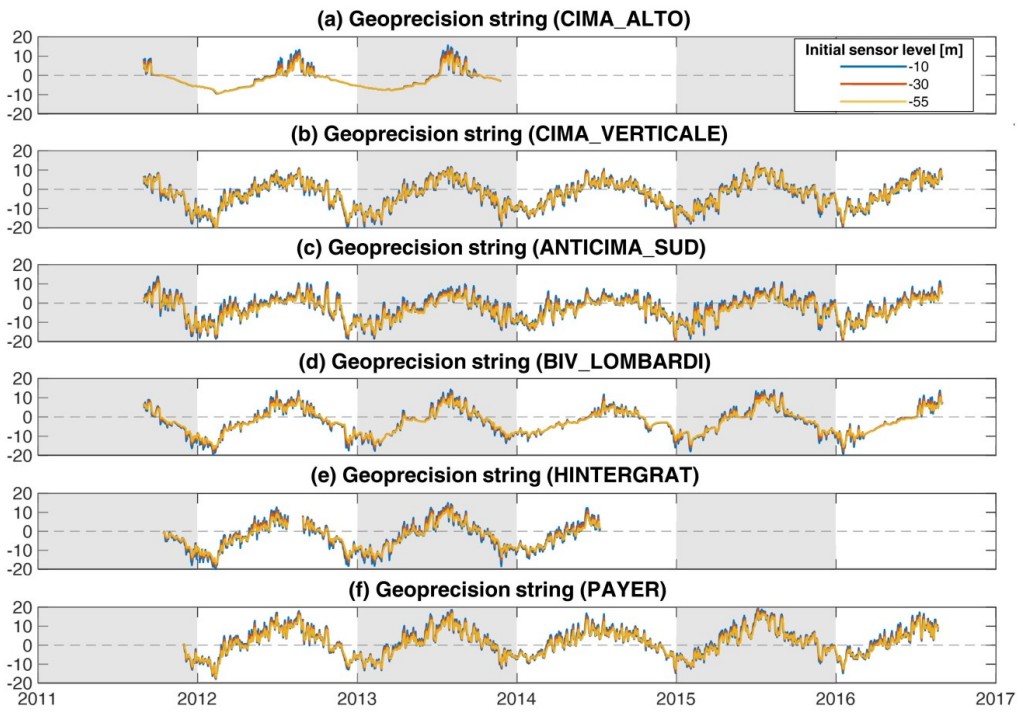


Figure 7. Rock wall temperature measurements at various sites and three depths on Mt. Ortles.

**4. Data quality control and assessment**
The temperature datasets presented in this work were carefully inspected to detect possible problems affecting
raw measurements and to ensure the highest possible accuracy. Data quality controls allowed assigning a
quality flag to each temperature record, as described in Table B1.
Air temperature sensors were exposed to harsh conditions, without protection from snow, rock or firn/ice as in
the case of sensors used to measure the englacial, soil and rock wall temperature. For this reason, the air
temperature sensors were subjected to possible damage by strong winds and lightning, ice and snow accretion,
and burial by snow in case of abundant snowfall. In addition, they were subjected to the typical issues affecting
air temperature measurements, arising from low wind speed and high solar radiation, worsened by high surface
albedo, which generally lead to errors due to heating during daytime (WMO, 2021). Sensor drifting should
also be taken into account as a potential problem.



In order to spot problematic periods we carried out a sensor-by-sensor intercomparison, calculating hourly and
daily temperature differences among pairs of sensors, including data from two neighbouring weather stations
(Madriccio, 2825 m a.s.l., and Cima Beltovo, 3328 m a.s.l., Weather Service - Autonomous Province of
Bozen/Bolzano) for additional confirmation. We compared temperature difference series with maintenance
logs, to understand the sources of malfunctions and anomalies, and to assign data quality flags to air
temperature series. Data recorded during malfunctions were handled as data gaps and removed from the
published series. Anomalies in periods of heavy snowfalls, which caused snow/ice accretion and a rapid
decrease of sensor height, are flagged with a specific code (Table B1).
Englacial and rock wall temperature were checked in the same way by calculating hourly and daily differences
among sensors located at the same site, and checking irregularities (i.e. sudden jumps in temperature
differences) in combination with field observations during maintenances. We have detected no malfunctions,
but it is possible that after maintenance operations (detailed in Table D1 to D4) a short period of a few hours
or a few days was required to reach a new equilibrium at englacial temperature monitoring sites. We have
highlighted these maintenance operations with potential impacts on measured temperature using a data quality
code, reported in Table B1.
Soil surface temperature data displayed no obvious anomalies and were checked in the 'zero-curtain' phase,
that is the 0°C plateau during the snowmelt phase. Only ORTL_07 required a correction of measured
temperature in 2014 (offset applied = -0.35°C) and 2015 (offset applied = -1.1°C), to correct discrepancies
larger than sensor accuracies reported in Table 1. We have highlighted these adjustments with a quality flag in
the corresponding data files (Table B1).

**5. Data availability**
The datasets from this study are publicly available at https://doi.org/10.5281/zenodo.7879969 (Carturan et al.,
2023). The data files are stored in Microsoft Excel .xlsx format. Detailed information on the file content and
structure is reported in the Appendix of this manuscript (Tables A1, B1 and C1).





## 6. Summary of observations and research outlook

The datasets collected on Mt. Ortles enable description of its thermal state within a time window of six years (2011-2016). This period is long enough to provide a picture of modern average conditions and interannual variability, and as such, it is useful as a baseline for possible future studies aimed at detecting changes and trends in monitored variables.

In the period from 02-10-2011 to 01-09-2016, the mean daily air temperature ranged between -30.1°C (12-02-2012) and 6.1°C (03-08-2013), averaging -8.3°C. These statistics have been extracted from a merged time series, which combines the sensors that have the longest time coverage (Fig. 2), i.e. the GEMINI TGP-4020 with natural ventilation before 15-06-2015 and the ONSET Hobo H8 (natural ventilation) from 15-06-2015. The air temperature reached hourly extremes of -33.3°C (09-02-2012 at 24:00 UTC) and 10.1°C (20-08-2012 at 10:00 UTC). These extremes must be viewed with caution due to the high sensitivity of short-term temperature fluctuations to possible errors, mainly due to low natural ventilation. The aspirated shield installed at the weather station proved subject to damage and malfunction. However, it was operational at the time when the two extreme values were recorded, providing identical minimum temperature of -33.3°C, and a maximum of 8.0°C at 13:00 of 20-08-2012. In the common period of operation, the average difference in mean daily air temperature among pairs of installed sensors did not exceed 0.60°C in absolute value (Table 2).

Table 2. Average difference in mean daily air temperature among pairs of installed sensors for air temperature measurements (f.a. = fan-aspirated; n.v. = natural ventilation). Differences are calculated in the common working period for pairs of sensors, i.e. they refer to different periods (overlaps are shown in Fig. 4). Column headings represent the first term of the difference calculation.

| | | Vaisala HMP155A (f.a.) | Vaisala HMP155A (n.v.) | Gemini TGP-4020 (f.a.) | Gemini TGP-4020 (n.v.) | Onset Hobo H8 Pro Temp (n.v.) |
|---|---|---|---|---|---|---|
| Vaisala (f.a.) | HMP155A | | -0.28 | 0.11 | 0.18 | -0.02 |
| Vaisala (n.v.) | HMP155A | | | 0.41 | 0.60 | 0.32 |
| Gemini (f.a.) | TGP-4020 | | | | 0.08 | -0.15 |

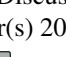



| | | | | | | | Gemini TGP-4020 (n.v.) | | | | -0.28 |
|---|---|---|---|---|---|---|---|---|---|---|---|



The mean daily air temperature was above 0°C for 39 days in 2012, 48 in 2013, 15 in 2014, 44 in 2015, and
31 in 2016. These results highlight a significant interannual variability in the length of the melt season at this
high-elevation site. The mean annual air temperature averaged -8.3°C, ranging between -8.6°C in 2012 and
2013, and -7.8°C in 2016 (Table 3).

Table 3. Mean annual air temperature recorded at the automatic weather station on Mt. Ortles, with five
different sensors (f.a. = fan-aspirated; n.v. = natural ventilation). The mean temperature is reported only for
years with less than one month missing of data, and is calculated between the 1st of September and the 31st of
August.

| Sensor | Vaisala HMP155A (f.a.) | Vaisala HMP155A (n.v.) | Gemini TGP-4020 (f.a.) | Gemini TGP-4020 (n.v.) | Onset Hobo H8 Pro Temp (n.v.) | Merged (n.v.) |
|---|---|---|---|---|---|---|
| Year | | | | | | |
| 2012 | -8.9 | | -8.7 | -8.6 | | -8.6 |
| 2013 | | | | -8.6 | -9.0 | -8.6 |
| 2014 | -8.4 | | -8.2 | -8.1 | -8.4 | -8.1 |
| 2015 | | | | | -8.6 | -8.3 |
| 2016 | | | | | -8.1 | -7.8 |


Englacial temperature measurements reveal warm firn and isothermal summer conditions down to a depth of
25 m on the upper part of the Alto dell'Ortles Glacier. The summer was cold (and snowy) enough only in 2014
to preserve below-zero temperature in firn and snow down to a depth of about 15 m at the AWS and 10 m at
the drilling site. On the other hand deep borehole data available until 10/04/2013 confirm that glacier ice below
the firn-ice transition is cold throughout the year, as detected during ice coring operations at the drilling site in
October 2011 (Gabrielli et al. 2012). The ice temperature decreases with depth reaching a minimum of -3°C
at the glacier bed, at 75 m below the surface, and does not change significantly throughout the year.



The Hintergrat glacier is also composed of cold ice, which is subject to net surface ablation at the string
installation site. Indeed, the sensor at 1.5 m initial depth was exposed at the surface in August 2013. A 1.2 m
layer of firn formed in 2014, but underwent complete ablation by the end of the following summer.
Soil surface temperature measurements, and in particular the mean annual ground surface temperature
(MAGST, Table 4), suggest the existence of permafrost on most of the monitored sites (Guglielmin et al.,
2003; Ballantyne, 2018), with the exception of the ORTL07 site which is at 2994 m a.s.l, close to the Payer
refuge (Fig. 1).   The results of ORTL_10 and ORTL_11 can be compared to analogous observations
(unpublished) carried out in the same period on Mt Vioz (3520 m a.s.l., 14 km south of Mt. Ortles), using the
same devices and field techniques at two sites with similar elevation and exposure. On Mt. Vioz the MAGST
was -2.1°C for the site with southern exposure and -2.9°C for the site with eastern exposure, indicating slightly
colder soil surface thermal conditions.

Table 4. Mean annual ground surface temperature (MAGST) recorded at six different sites on Mt. Ortles. Site
locations are reported in Fig. 1. MAGST is reported only for years with less than one month of missing data,
and is calculated between the 1st of September and the 31st of August.

| Sensor | ORTL_05 | ORTL_06 | ORTL_07 | ORTL_08 | ORTL_10 | ORTL_11 |
|--------|---------|---------|---------|---------|---------|---------|
| Year |  |  |  |  |  |  |
| 2011 | -2.6 | -2.4 | 1.3 | -1.2 | -1.2 | -2.4 |
| 2012 |  |  |  |  |  |  |
| 2013 | -3.5 | -2.7 | 1.1 | -1.0 | -0.7 | -2.1 |
| 2014 | -3.2 | -2.1 | 0.9 | -0.5 | -1.5 |  |
| 2015 |  |  | 1.9 | -0.3 | -1.0 |  |
| 2016 | -3.4 | -1.6 | 0.7 | -1.6 | -1.9 |  |




Rock wall temperature provided results that are in line with soil surface temperature measurements. The
warmest site was close to the Payer refuge, with mean annual rock surface temperature (MARST) above the
freezing level (Table 5). All the other monitored rock walls displayed below-freezing MARST and similar
behaviour, with the exception of CIMA_ALTO, close to the Mt. Ortles summit, where rock temperature
fluctuations appear to be dampened by snow accumulation between September and May (Fig. 7).



The collected data are being analysed for the interpretation of the ice core drilled in the framework of the Ortles
Project. In particular, air and englacial temperature data are used for developing and validating a model that
aims at reproducing the formation of the isotopic record in snow and firn.
Together with rock wall and soil surface temperature, these datasets represent unique observations at such
elevation in the eastern European Alps, and may contribute to the study and understanding of specific aspects
of the climatic sensitivity of the alpine cryosphere. For example, they can be used for the development of
permafrost distribution and degradation models, air temperature simulations over glacierized areas (including
the so-called glacier cooling effect), snow and glacier mass balance models, glacio-hydrological forecasting
systems, or dynamic glacier models that take into account the thermal state of glaciers and its variability.

Table 5. Mean annual rock surface temperature (MARST) recorded at six different site on Mt. Ortles. Site
locations are reported in Fig. 1. MARST is reported only for years with less than one month of missing data,
and is calculated between the 1st of September and the 31st of August.

| LOCATION | CIMA_ALTO | | | CIMA_VERTICALE | | | ANTICIMA_SUD | | | BIVACCO_LOMBARDI | | | HINTERGRAT | | | PAYER | | |
|---|---|---|---|---|---|---|---|---|---|---|---|---|---|---|---|---|---|---|
| Depth (m) | 0.10 | 0.30 | 0.55 | 0.10 | 0.30 | 0.55 | 0.10 | 0.30 | 0.55 | 0.10 | 0.30 | 0.55 | 0.10 | 0.30 | 0.55 | 0.10 | 0.30 | 0.55 |
| Year | | | | | | | | | | | | | | | | | | |
| 2012 | -2.1 | -2.3 | -2.6 | -2.6 | -2.5 | -2.5 | -1.8 | -2.2 | -2.7 | -2.4 | -2.3 | -2.4 | | | | | | |
| 2013 | -2.0 | -2.3 | -2.5 | -3.1 | -3.1 | -3.0 | -3.1 | -3.5 | -3.9 | -2.8 | -2.7 | -2.7 | -2.2 | -2.1 | -2.2 | 1.4 | 1.2 | 0.9 |
| 2014 | | | | -2.9 | -2.8 | -2.8 | -2.5 | -2.8 | -3.3 | -2.8 | -2.7 | -2.7 | | | | 1.8 | 1.5 | 1.2 |
| 2015 | | | | -2.1 | -2.1 | -2.1 | -2.2 | -2.6 | -3.1 | -1.5 | -1.4 | -1.6 | | | | 2.7 | 2.4 | 2.1 |
| 2016 | | | | -2.4 | -2.3 | -2.3 | -2.2 | -2.5 | -3.0 | -3.2 | -3.1 | -3.1 | | | | 1.9 | 1.6 | 1.3 |




# APPENDICES

## APPENDIX A: Variables in data files

Table A1. Column names for variables reported in data files.

| Variable | Column name |
|---|---|
| Air temperature (fan-aspirated Vaisala HMP155A) | Air_T_HMP_Asp |
| Air temperature (natural-ventilation Vaisala HMP155A) | Air_T_HMP_Nat |
| Air temperature (fan-aspirated Gemini TGP-4020) | Air_T_TGP_Asp |
| Air temperature (natural-ventilation Gemini TGP-4020) | Air_T_TGP_Nat |
| Air temperature (natural-ventilation Onset Hobo H8 Pro Temp) | Air_T_H8_Nat |
| Englacial temperature at the AWS site | AWS_En (depth m) |
| Englacial temperature at the borehole drilling site (short string) | BH_En_SS (depth m) |
| Englacial temperature at the borehole drilling site (long string) | BH_En_LS (depth m) |
| Englacial temperature at the Hintergrat Glacier | HG_En (depth m) |
| Soil surface temperature at bivouac Lombardi - ORTL_05 | GST_ORTL05 |
| Soil surface temperature at bivouac Lombardi - ORTL_06 | GST_ORTL06 |
| Soil surface temperature at refuge Payer - ORTL_07 | GST_ORTL07 |
| Soil surface temperature at refuge Payer - ORTL_08 | GST_ORTL08 |
| Soil surface temperature at Hintergrat ridge - ORTL_10 | GST_ORTL10 |
| Soil surface temperature at Hintergrat ridge - ORTL_11 | GST_ORTL11 |
| Rock wall temperature at Mt. Ortles summit - CIMA_ALTO | Rw_ALTO (depth m) |
| Rock wall temperature Mt. Ortles summit - CIMA_VERTICALE | Rw_VERTICALE (depth m) |
| Rock wall temperature at Vorgipfel - ANTICIMA_SUD | Rw_ANTICIMA (depth m) |
| Rock wall temperature at bivouac Lombardi - BIV_LOMBARDI | Rw_LOMBARDI (depth m) |
| Rock wall temperature at Hintergrat - HINTERGRAT | Rw_HINTERGRAT (depth m) |
| Rock wall temperature at refuge Payer - PAYER | Rw_PAYER (depth m) |




**APPENDIX B: Quality flags for data files**


Table B1. Quality code flags reported in data files, their meaning and explanations.


| Quality code flag ("_Fl" inflection in column names) | Meaning | Explanation |
|---|---|---|
| 1 | Good data | No issues detected during quality checks |
| 0 | No data | Data missing or removed (malfunctioning, physically implausible, sensor/device damaged, sensor underneath snow) |
| 2 | Maintenance | Data are affected by field maintenance of instrumentation |
| 3 | Ice/snow accretion | The air temperature data are affected by ice or snow accretion |
| 4 | Small height of the sensor | The air temperature sensor is less than 1 m above the snow surface |
| 5 (offset) | Sensor offset | Offset applied to correct soil surface temperature data, based on the zero-curtain phase during snow melt (offset value in brackets) |




**APPENDIX C: Data files structure**


Table C1. Structure of data files. For sensors at different depth below the surface, the depth in m is reported


after the variable name, in brackets.


| Date and hour (UTC) | Variable name (depth m) | Quality flag code |
|---|---|---|
| DD/MM/YYYY HH:MM | value | code |



**APPENDIX D: Maintenance logs**


Table D1. Field operations and maintenance for the air temperature sensors mounted at the AWS on Mt. Ortles.


| Date | Field operations |
|---|---|
| 01/10/2011 | AWS setup and datalogger launch |
| 18/06/2012 | Datalogger download, check of sensor status and functioning |
| 07/09/2012 | Datalogger download, check of sensor status and functioning, 2 m increase in height of support tower, installation of two additional sensors (Vaisala HMP155A with natural ventilation, and Onset Hobo H8 Pro Temp) |
| 01/07/2013 | Datalogger download, check of sensor status and functioning. The fan-aspirated radiation shield was found damaged and not working |





| 03/09/2013 | Datalogger download, check of sensor status and functioning, 2 m increase in height of support tower. The fan-aspirated radiation shield was repaired and resumed working properly |
| --- | --- |
| 03/07/2014 | Datalogger download, check of sensor status and functioning |
| 23/09/2014 | 2 m elongation of support tower. Sensors have been partially buried by snow between 2 and 23/09/2014 |
| 29/06/2015 | Datalogger download, check of sensor status and functioning. Support tower lengthened by 2 m. Sensors have been partially buried by snow between 15 and 29/06/2015. Removal of all sensors except the Onset Hobo H8 Pro Temp |
| 31/08/2015 | Onset Hobo H8 Pro Temp download, check of sensor status and functioning |
| 02/09/2016 | Onset Hobo H8 Pro Temp download, check of sensor status and functioning. Sensor removed |



Table D2. Field operations and maintenance for the englacial temperature sensors installed on Mt. Ortles.

| Date | Field operations |
| --- | --- |
| 17/11/2011 | Installation and launch of the Sommer thermistor strings at the borehole drilling site and at the Hintergrat Glacier. |
| 18/06/2012 | Installation and launch of the Geoprecision thermistor string at the AWS site. Download and maintenance (battery replacement) of the Sommer thermistor strings at the borehole drilling site |
| 28/08/2012 | Download and maintenance (battery replacement) of the Sommer thermistor string at the Hintergrat Glacier. Sensor at 7.5 m initial depth stopped working on 18/08/2012 |
| 07/09/2012 | Download of the Geoprecision thermistor string at the AWS site. Download and maintenance (battery replacement) of the Sommer thermistor strings at the borehole drilling site |
| 01/07/2013 | Download of the Geoprecision thermistor string at the AWS site. Download and maintenance (battery replacement, logger raised to the surface) of the Sommer thermistor strings at the borehole drilling site |
| 23/08/2013 | Download and maintenance (battery replacement) of the Sommer thermistor string at the Hintergrat Glacier. The sensor at 1.5 m initial depth was above the glacier surface |
| 03/09/2013 | Download of the Geoprecision thermistor string at the AWS site. Download and maintenance (battery replacement) of the Sommer thermistor strings at the borehole drilling site. Long string stopped working on 10/04/2013 |
| 03/07/2014 | Download and maintenance (battery replacement, logger replacement) of the short Sommer thermistor string at borehole drilling site. Removal of the datalogger of the long Sommer thermistor string at borehole drilling site (no longer working). |
| 28/08/2014 | Download and maintenance (battery replacement) of the Sommer thermistor string at the Hintergrat Glacier. Sensor at 5.5 m initial depth stopped working on 17/08/2014 |
| 23/09/2014 | Download and maintenance (battery replacement, logger raised to the surface) of the short Sommer thermistor string at the borehole drilling site |
| 18/10/2014 | Download of the Geoprecision thermistor string at the AWS site |
| 29/06/2015 | Download of the Geoprecision thermistor string at the AWS site |
| 27/08/2015 | Download and maintenance (battery replacement) of the Sommer thermistor string at the Hintergrat Glacier |
| 31/08/2015 | Download and removal of the short Sommer thermistor string at the borehole drilling site |
| 23/08/2016 | Download and removal of the Sommer thermistor string at the Hintergrat Glacier |
| 02/09/2016 | Download of the Geoprecision thermistor string at the AWS site |





Table D3. Field operations and maintenance for the soil surface temperature sensors installed on Mt. Ortles.

| Date | Field operations |
|---|---|
| 02/09/2010 | Installation and launch of the ORTL_05, ORTL_06, ORTL_07 and ORTL_08 Gemini TGP-4020 |
| 23/09/2010 | Installation and launch of the ORTL_10 and ORTL_11 Gemini TGP-4020 |
| 28/08/2012 | Download and maintenance (battery replacement, logger re-launch) at the ORTL_07, ORTL_08, ORTL_10 and ORTL_11 |
| 07/09/2012 | Download and maintenance (battery replacement, logger re-launch) at the ORTL_05 and ORTL_06 |
| 23/08/2013 | Download and maintenance (battery replacement, logger re-launch) at the ORTL_07, ORTL_08, ORTL_10 and ORTL_11 |
| 03/09/2013 | Download and maintenance (battery replacement, logger re-launch) at the ORTL_05 and ORTL_06 |
| 28/08/2014 | Download and maintenance (battery replacement, logger re-launch) at the ORTL_07, ORTL_08 and ORTL_10. ORTL_05, ORTL_06 and ORTL_11 not found, buried by snow |
| 27/08/2015 | Download and maintenance (battery replacement, logger re-launch) at the ORTL_07, ORTL_08 and ORTL_10 |
| 31/08/2015 | Download and maintenance (battery replacement, logger re-launch) at the ORTL_05 and ORTL_06 |
| 23/08/2016 | Download and removal of the ORTL_07, ORTL_08 and ORTL_10 |
| 02/09/2016 | Download and removal of the ORTL_05 and ORTL_06 |


Table D4. Field operations and maintenance for the rock wall temperature sensors installed on Mt. Ortles.

| Date | Field operations |
|---|---|
| 30/08/2011 | Installation and launch of the Geoprecision thermistor strings at CIMA_ALTO, CIMA_VERTICALE, ANTICIMA_SUD, BIV_LOMBARDI |
| 14/10/2011 | Installation and launch of the Geoprecision thermistor string at HINTERGRAT |
| 28/11/2011 | Installation and launch of the Geoprecision thermistor string at PAYER |
| 28/08/2012 | Download of thermistor strings at HINTERGRAT and PAYER. Repair of the Geoprecision thermistor string at HINTERGRAT |
| 07/09/2012 | Download of thermistor strings at CIMA_ALTO, CIMA_VERTICALE, ANTICIMA_SUD, BIV_LOMBARDI |
| 23/08/2013 | Download of thermistor strings at HINTERGRAT and PAYER |
| 03/09/2013 | Download of thermistor strings at CIMA_ALTO, CIMA_VERTICALE, ANTICIMA_SUD, BIV_LOMBARDI |
| 28/08/2014 | Download of thermistor string at PAYER |
| 01/09/2014 | Download of thermistor strings at HINTERGRAT, damaged, removed |
| 27/08/2015 | Download of thermistor string at PAYER |
| 31/08/2015 | Download of thermistor strings at CIMA_ALTO, CIMA_VERTICALE, ANTICIMA_SUD, BIV_LOMBARDI. CIMA_ALTO damaged, removed |
| 23/08/2016 | Download and removal of the thermistor string at PAYER |
| 02/09/2016 | Download and removal of the thermistor strings at CIMA_VERTICALE, ANTICIMA_SUD, BIV_LOMBARDI |




**APPENDIX E: Characteristics of measurement sites**

Table E1. Topographic and geomorphological characteristics of sites instrumented for temperature measurements.

| Measured variable | Elevation (m a.s.l.) | East coordinate UTM WGS84 (m) | North coordinate UTM WGS84 (m) | Aspect | Slope (degrees) | Site description |
|---|---|---|---|---|---|---|
| Air Temperature (automatic weather station) | 3830 | 618254 | 5151614 | NW | 11 | Upper accumulation area of Alto dell'Ortles Glacier |
| Snow and firn temperature at the AWS site | 3830 | 618260 | 5151619 | NW | 11 | Upper accumulation area of Alto dell'Ortles Glacier |
| Englacial temperature at the borehole drilling site (short and long strings) | 3859 | 618373 | 5151536 | W | 7 | Upper accumulation area of Alto dell'Ortles Glacier |
| Englacial temperature at the Hintergrat Glacier | 3476 | 619435 | 5151395 | N | 12 | Hintergrat Glacier |
| Soil surface temperature at Lombardi bivouac - ORTL_05 | 3351 | 618202 | 5152846 | SW | 7 | Northern ridge of Mt. Ortles, bedrock covered by a thin layer of debris (fine gravel, sand) |
| Soil surface temperature at Lombardi bivouac - ORTL_06 | 3371 | 618284 | 5152772 | N | 22 | Northern ridge of Mt. Ortles, recently deglaciated bedrock covered by a discontinuous layer of loose debris (fine gravel, sand) |
| Soil surface temperature at Payer - refuge ORTL_07 | 2994 | 618361 | 5153936 | N | 22 | Northern ridge of Mt. Ortles, bedrock covered by a thick layer of debris (pebbles, gravel, sand) with sparse vegetation |
| Soil surface temperature at Payer - refuge ORTL_08 | 2899 | 618287 | 5154105 | W | 36 | Northern ridge of Mt. Ortles, bedrock covered by coarse debris with isolated areas of thinner debris (fine sand and silt). |
| Soil surface temperature at Hintergrat ridge ORTL_10 | 3460 | 619628 | 5151341 | S | 22 | Eastern ridge of Mt. Ortles, bedrock covered by a layer of debris (fine gravel, sand). |
| Soil surface temperature at Hintergrat ridge ORTL_11 | 3466 | 619491 | 5151374 | SE | 11 | Eastern ridge of Mt. Ortles, bedrock covered by a thin layer of coarse debris (gravel, sand), close to the edge of the Hintergrat Glacier |
| Rock wall temperature at Mt. Ortles summit - CIMA_ALTO | 3900 | 618512 | 5151691 | E | 70 | 70 m south of Mt. Ortles summit (3905 m), in a sub-vertical rock face about 30 m below the crest edge |
| Rock wall temperature Mt. Ortles summit - CIMA_VERTICALE | 3880 | 618512 | 5151691 | E | 90 | 70 m south of Mt. Ortles summit (3905 m), in a vertical rock face about 50 m below the ridge, 20 m below CIMA_ALTO |
| Rock wall temperature at Vorgipfel - ANTICIMA_SUD | 3810 | 618327 | 5151269 | S | 90 | Vertical rock face, about 10 m below the upper rock wall edge |
| Rock wall temperature at Lombardi bivouac - BIV_LOMBARDI | 3351 | 618213 | 5152784 | W | 70 | Northern ridge of Mt. Ortles, sub-vertical rock wall, about 30 m below the crest edge |
| Rock wall temperature at Hintergrat - HINTERGRAT | 3370 | 619710 | 5151334 | NE | 90 | Eastern ridge of Mt. Ortles, vertical rock wall,about 10 m below the crest edge |
| Rock wall temperature at Payer refuge - PAYER | 3030 | 618372 | 5153812 | SE | 90 | Northern ridge of Mt. Ortles, vertical rock wall, about 20 m below the crest edge |


**APPENDIX F: Description of the measuring equipment for air temperature**


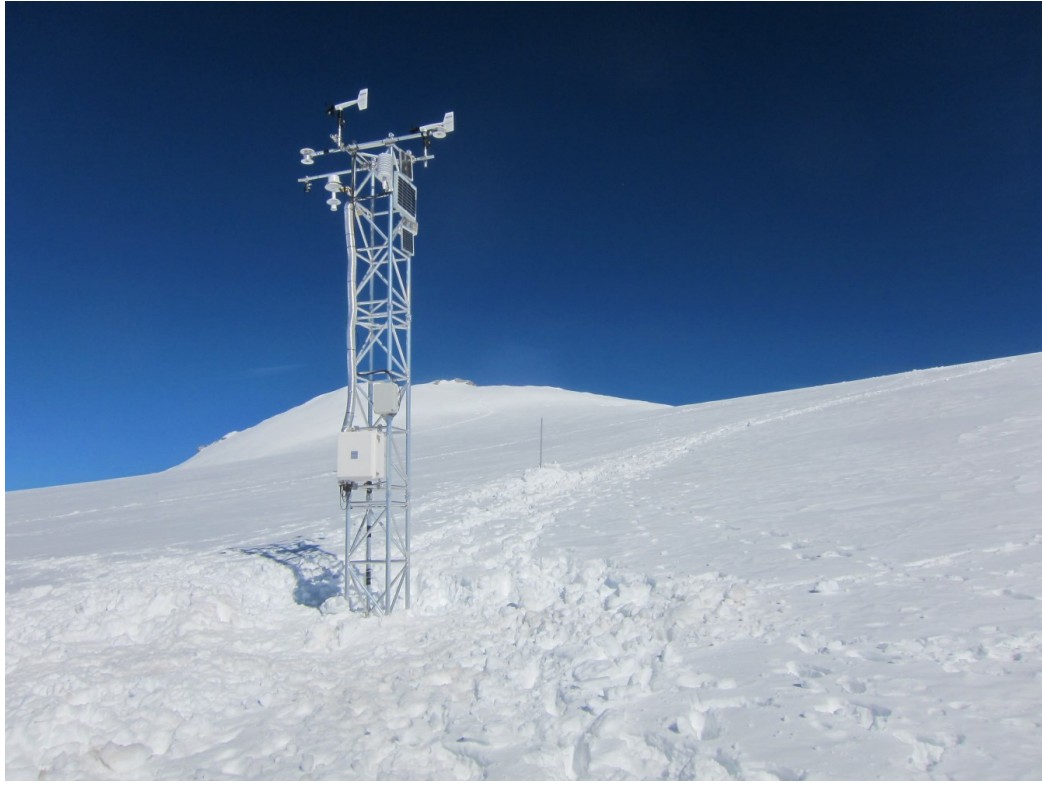


Figure F1. The automatic weather station (AWS) installed on Mt. Ortles, whose summit is visible in the

background. The stake behind the weather station indicates the site of the Geoprecision thermistor string. Photo

taken on 7 September 2012, after the lengthening of the support tower.


[Figure F2 photograph]


Figure F2. The wooden boards placed at the bottom of the support aluminum tower, at 2 m depth in the firn,

during the AWS installation. Photo taken on 30 September 2011.

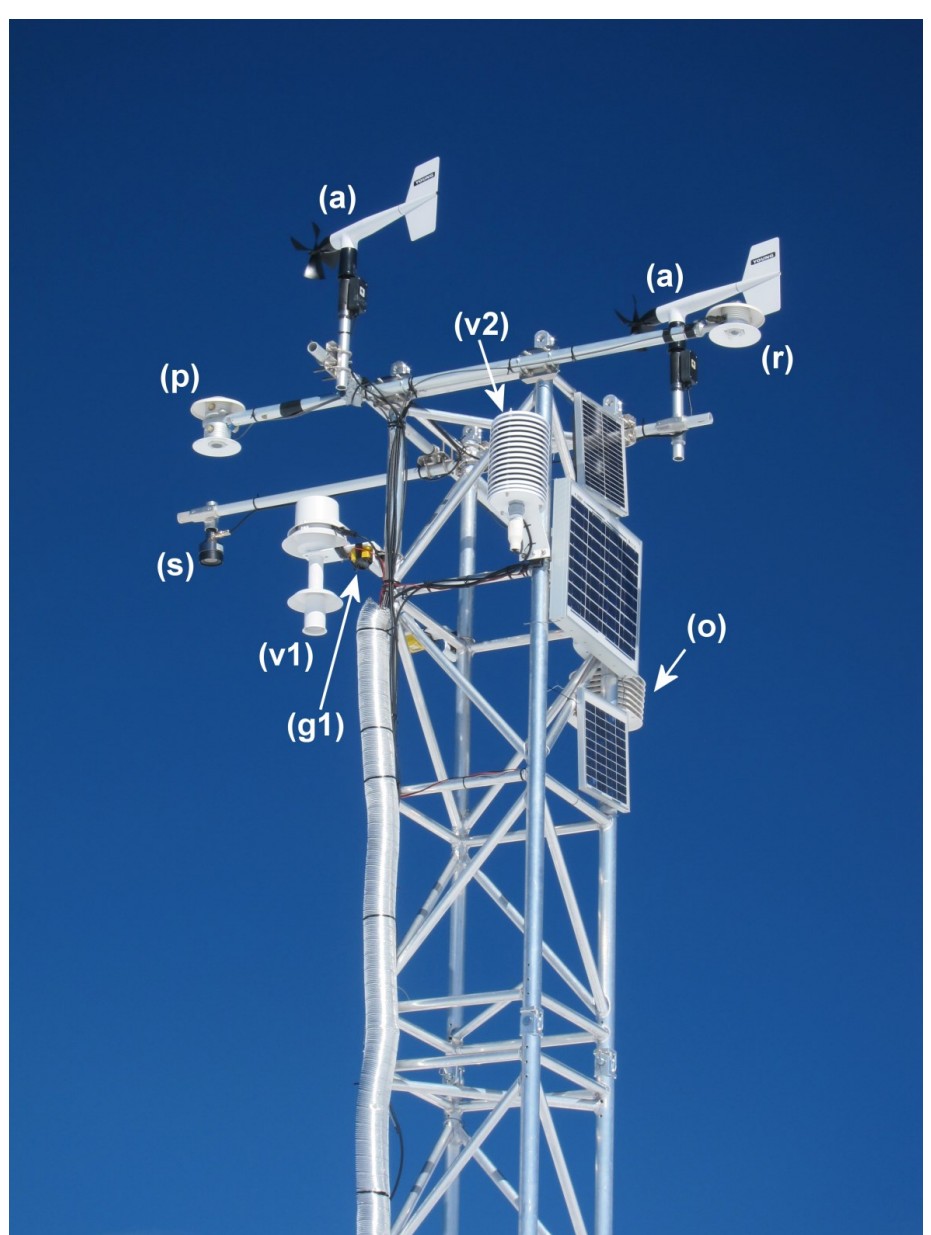


Figure F3. Detail of the AWS seen from the west: a) R. M. Young 05103 anemometers, p) Delta Ohm LP

PIRG 01 pyrgeometers, r) Delta Ohm LP Pyra 05 radiometers, s) Campbell Scientific SR50A snow depth

sensors, v1) Vaisala HMP155A inside the R. M. Young 43502 fan-aspirated radiation shield, g1) Gemini TGP-

4020 datalogger inside the R. M. Young 43502 fan-aspirated radiation shield, v2) Vaisala HMP155A inside

the 15-plates Campbell Scientific MET 21 radiation shield with natural ventilation, o) Onset Hobo H8 Pro



Temp datalogger inside the 8-plates Davis 7714 radiation shield with natural ventilation. Photo taken on 7
September 2012.


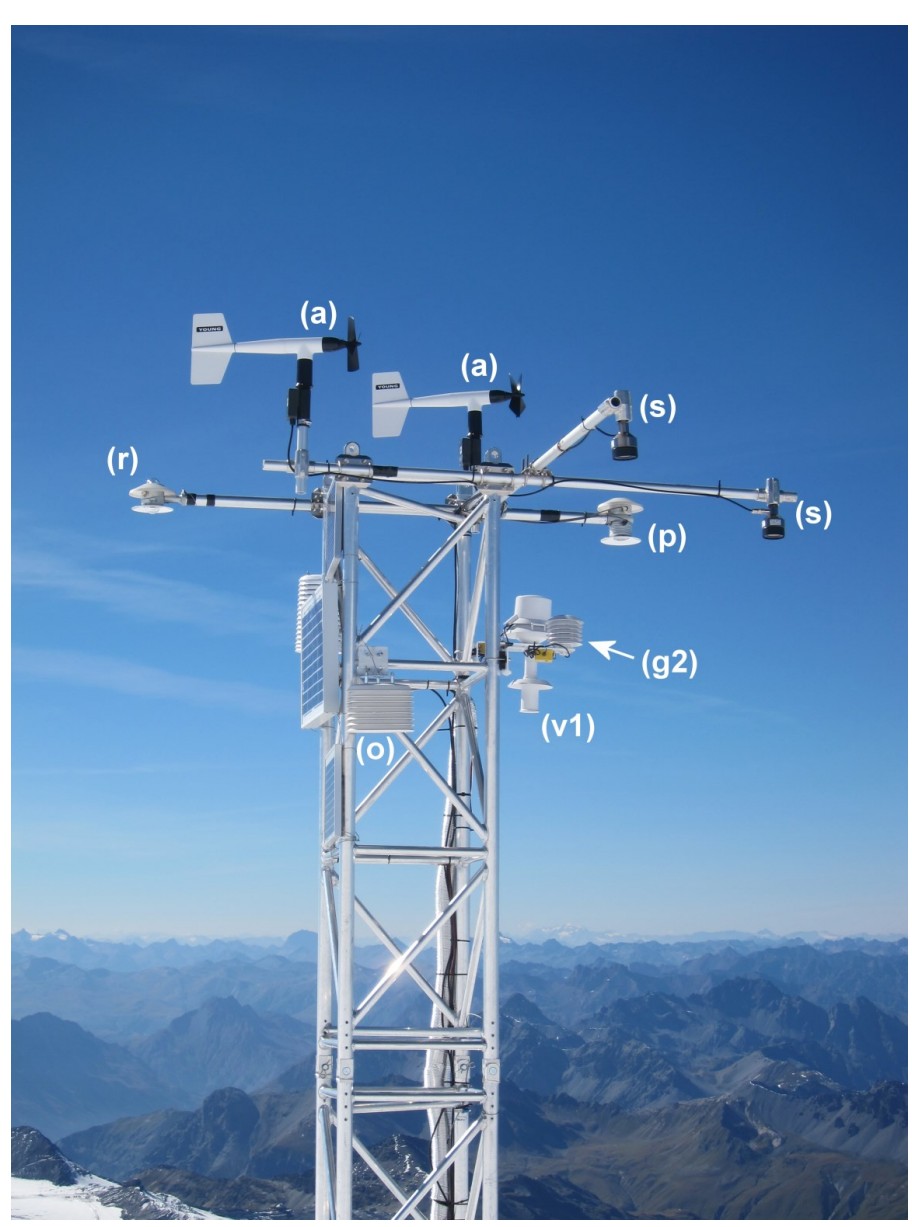

Figure F4. Detail of the AWS seen from the east: a) R. M. Young 05103 anemometers, p) Delta Ohm LP PIRG
01 pyrgeometers, r) Delta Ohm LP Pyra 05 radiometers, s) Campbell Scientific SR50A snow depth sensors,



v1) Vaisala HMP155A inside the R. M. Young 43502 fan-aspirated radiation shield, g2) Gemini TGP-4020
datalogger inside the 6-plates R.M. Young 41303-5 radiation shield with natural ventilation, o) Onset Hobo
H8 Pro Temp datalogger inside the 8-plates Davis 7714 radiation shield with natural ventilation. Photo taken
on 7 September 2012.




**APPENDIX G: Description of the measuring equipment for englacial temperature**

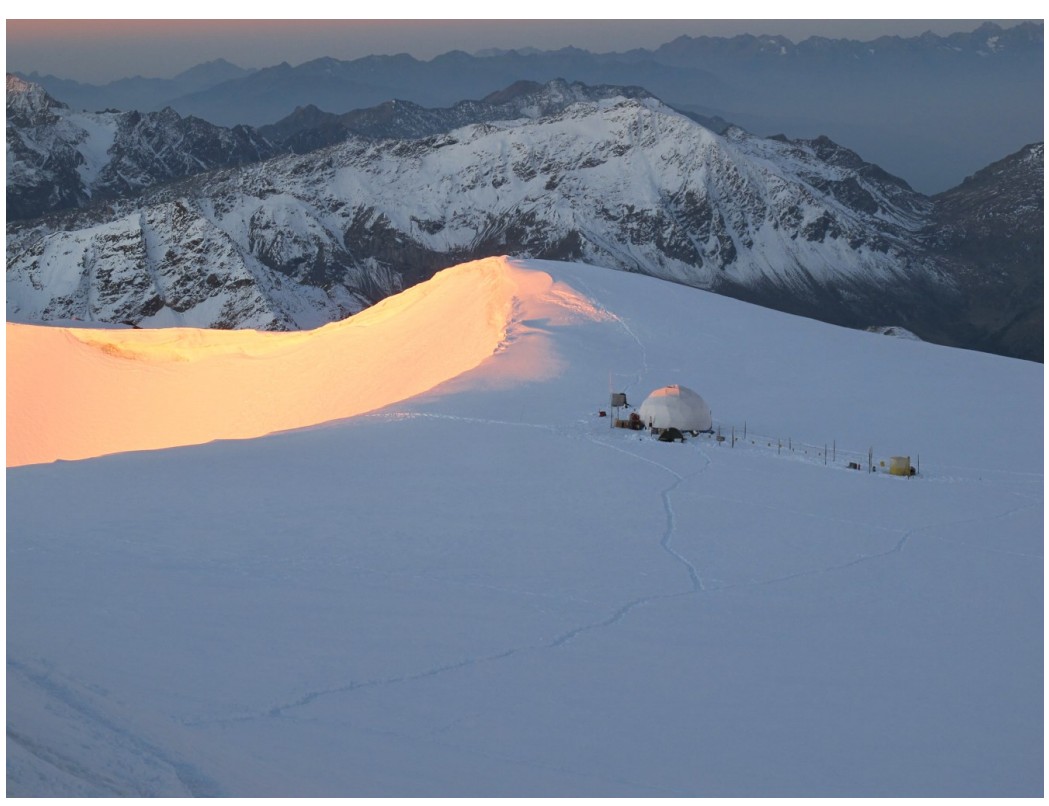


Figure G1. The drilling site seen from the summit of Mt. Ortles. The Vorgipfel-Anticima Sud is visible in the
background. Photo taken on 1 October 2011 during the ice drilling operations, and before setting up the drilling
site thermistor strings for englacial temperature measurements.








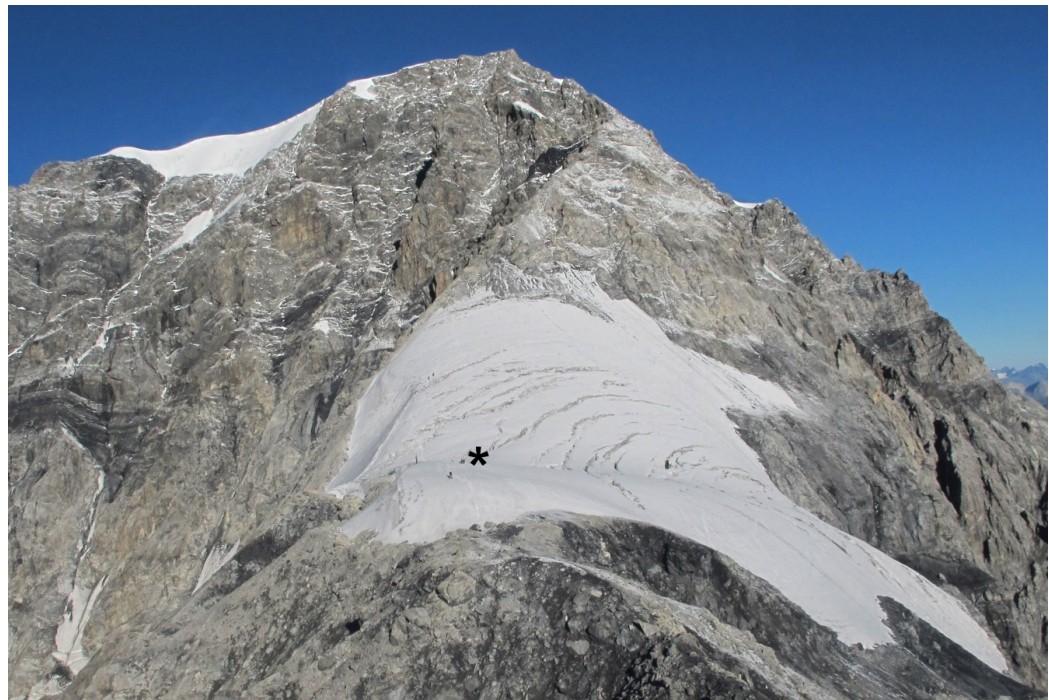

Figure G2. The Hintergrat Glacier seen from the south east (aerial photo taken on 28 August 2012). The black
asterisk indicates the location of the borehole equipped with the thermistor string.








Earth System
Science
Data


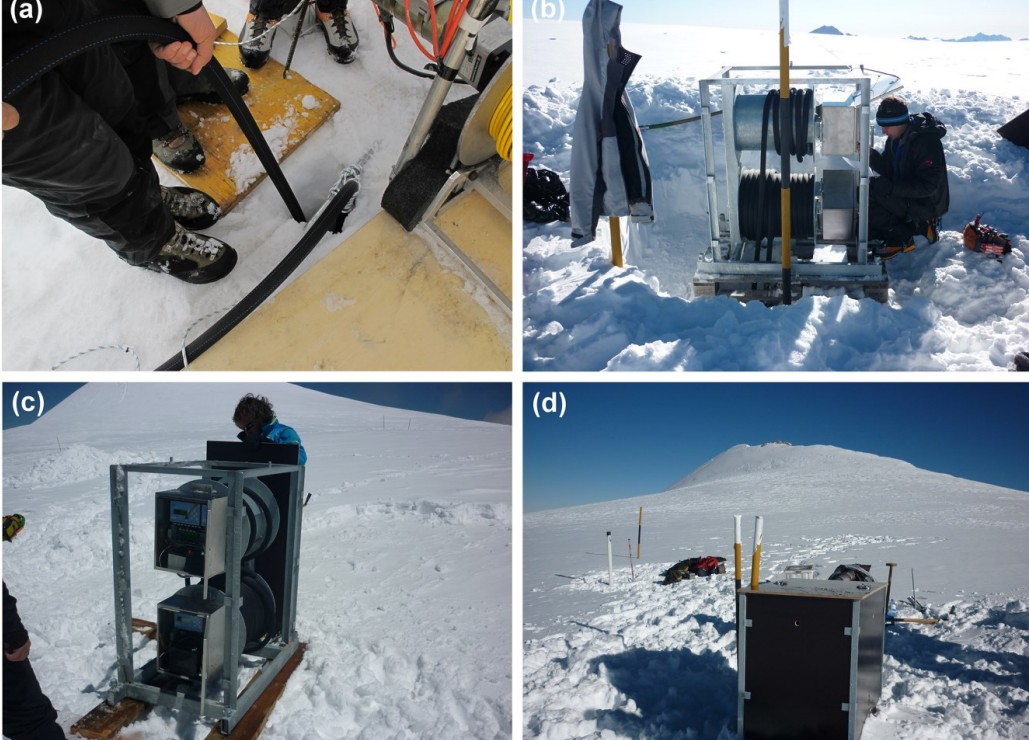

Figure G3. a) Lowering of a thermistor string into the borehole n. 3 at the drilling site. b) The winding systems
of the two thermistor strings installed at the drilling site. c) The two metal boxes containing the thermistor
string data loggers and the batteries. d) Final arrangement of the box housing the winding systems and the data
loggers. The summit of Mt. Ortles is visible in the background. Photos taken on 17 November 2011.









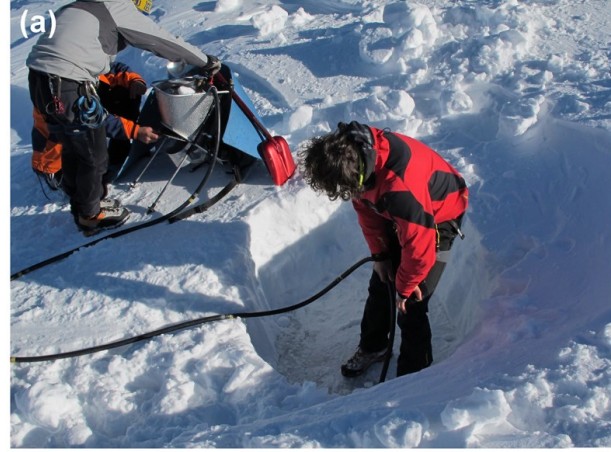

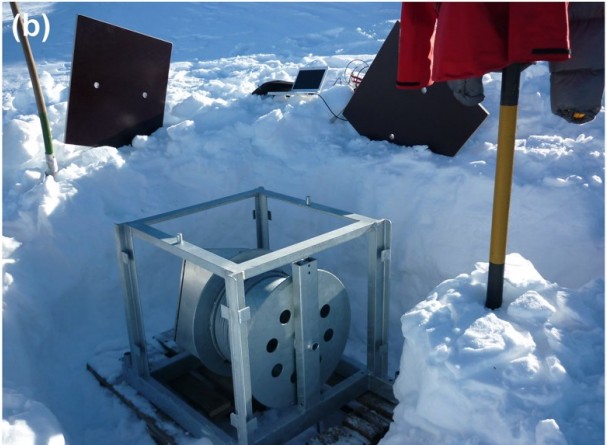

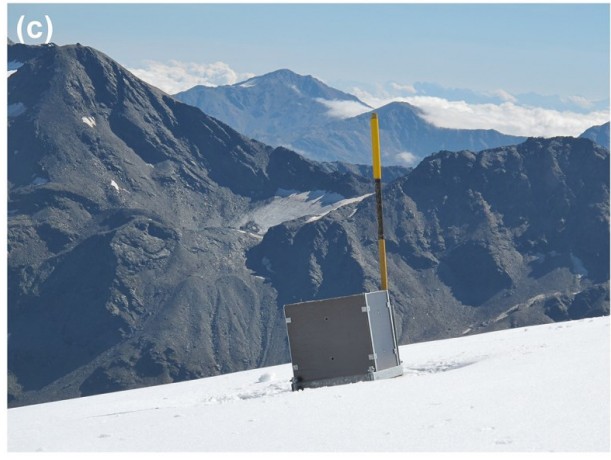


Figure G4. a) Borehole drilling at the Hintergrat Glacier using a steam ice drill. b) The box containing the
winding system, the thermistor string data logger and the batteries. c) Final arrangement of the box housing
the winding systems and the data logger. Photos taken on 17 November 2011 and 27 August 2015.

**APPENDIX H: Description of the measuring equipment for soil surface temperature**

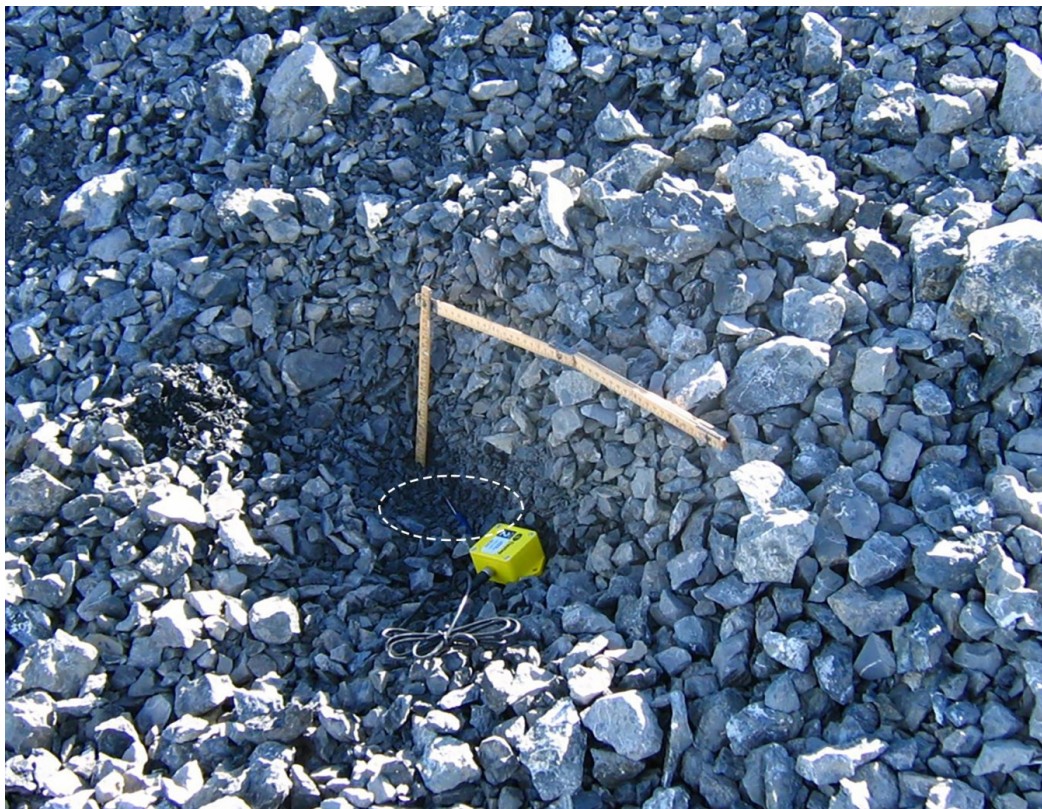

Figure H1. The soil surface temperature datalogger (Gemini TGP-4020) installed at the ORTL_05 site, close
to the Lombardi bivouac. The white ellipse indicates the PB-5001 external probe placed underneath the debris
surface. Photo taken on September 2010.





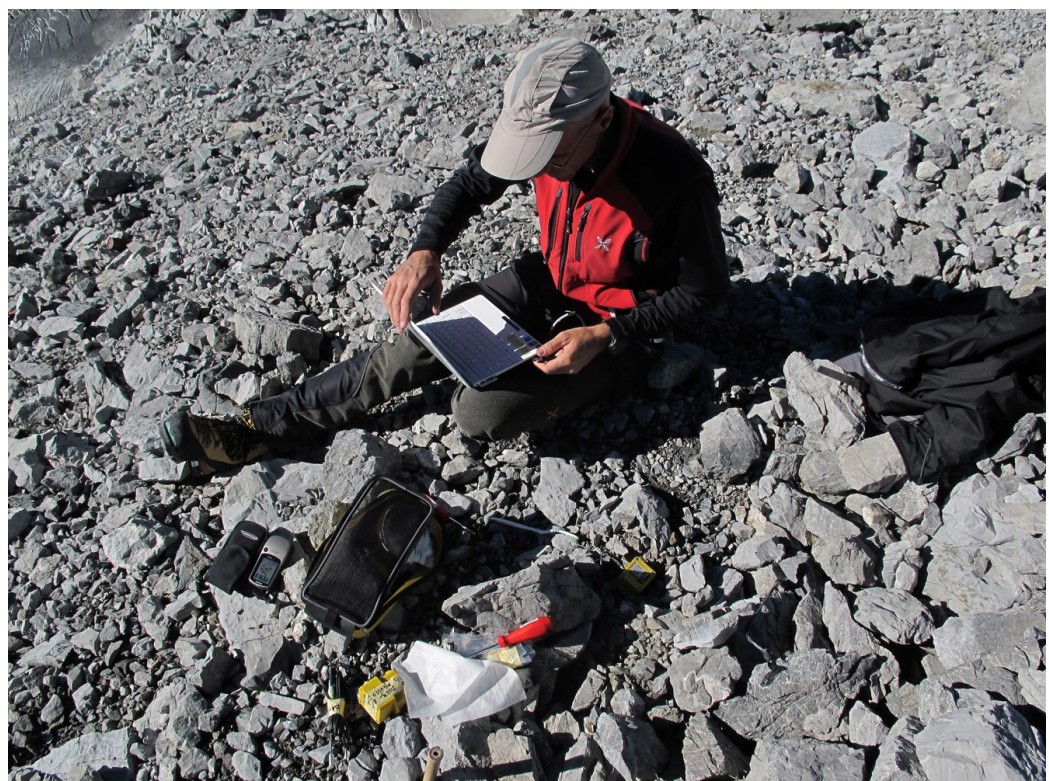


Figure H2. Data download and logger maintenance at the ORTL_10 soil surface temperature site on 28
August 2012.





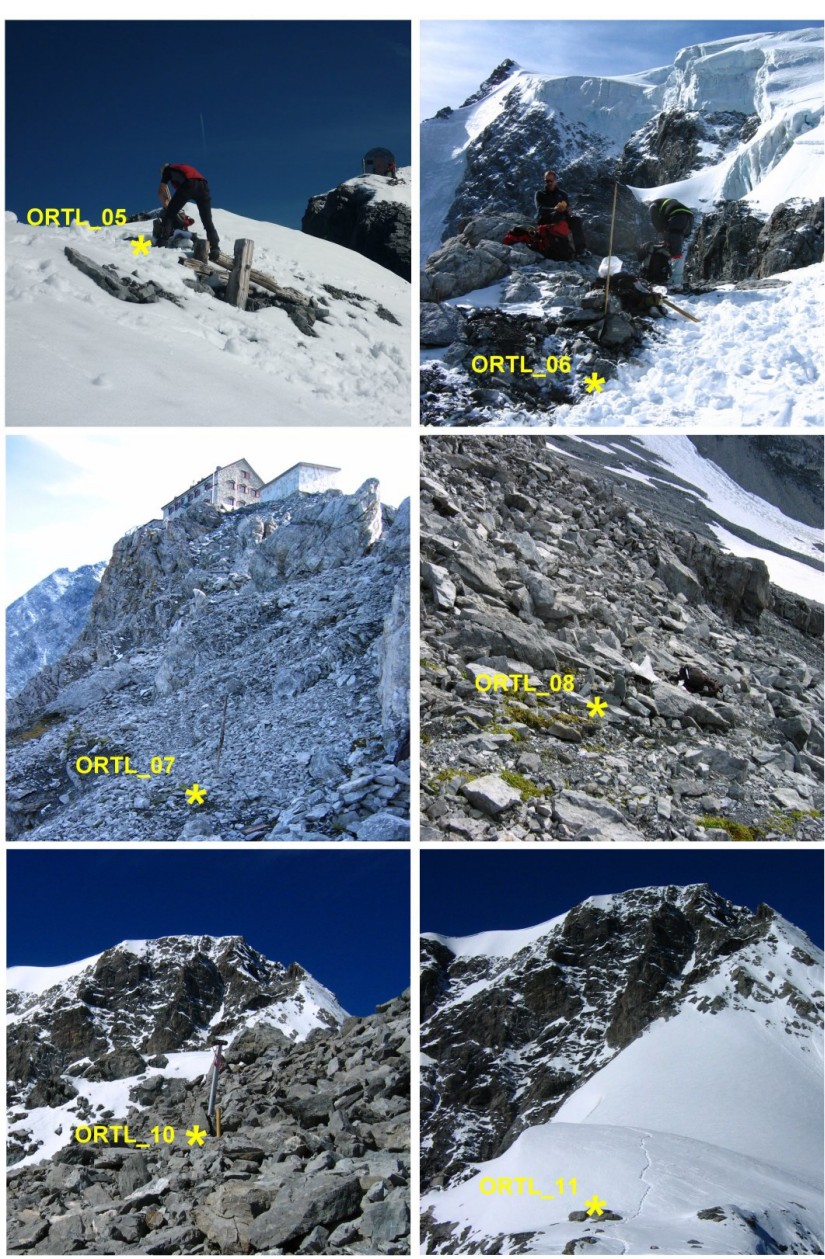

Figure H3. Location of the six sites equipped with dataloggers for soil surface temperature measurement on
Mt. Ortles.



**APPENDIX I: Description of the measuring equipment for rock wall temperature**


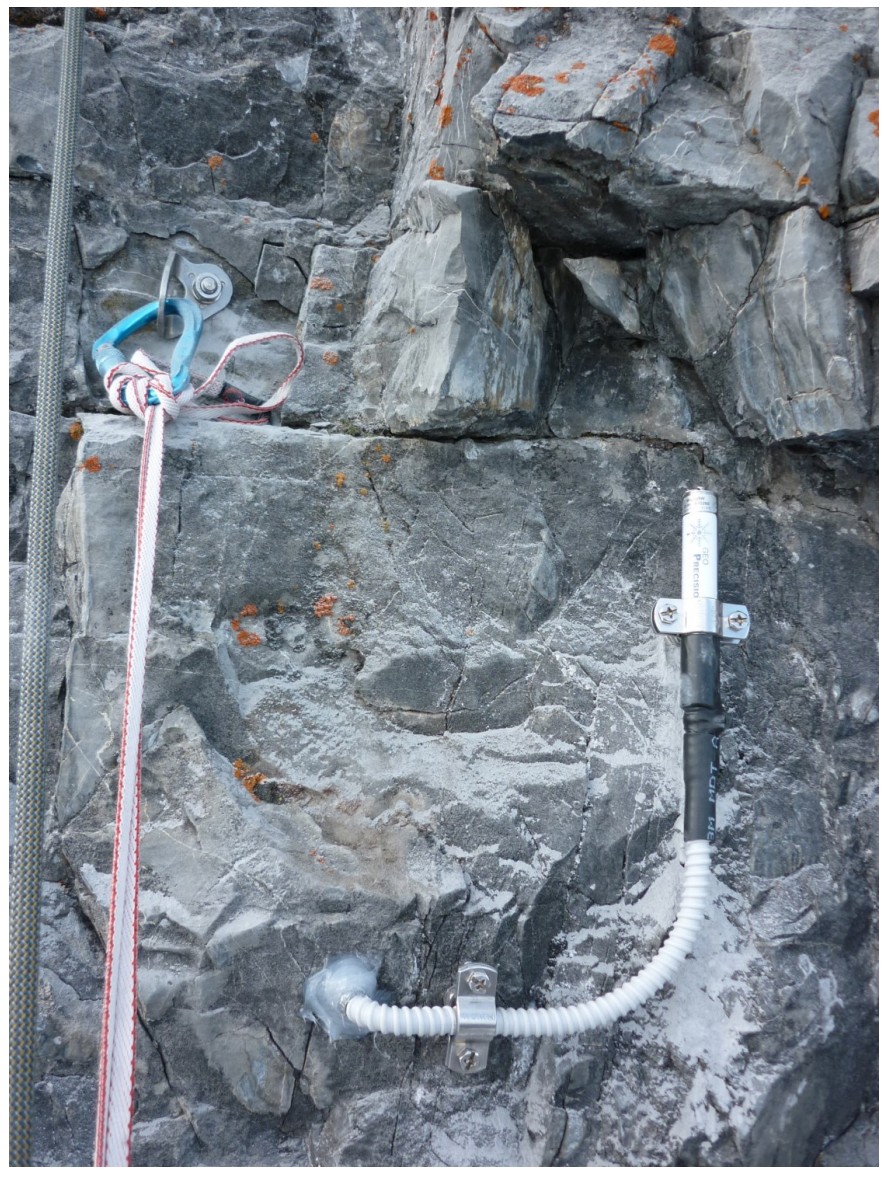


Figure I1. The rock wall temperature datalogger (Geoprecision thermistor string) installed close to the Payer

refuge. The datalogger is anchored to the rock wall and is connected to three temperature sensors placed at 0.1,

0.3 and 0.55 m depth inside a horizontal hole drilled in the rock wall. Photo taken on 28 November 2011.


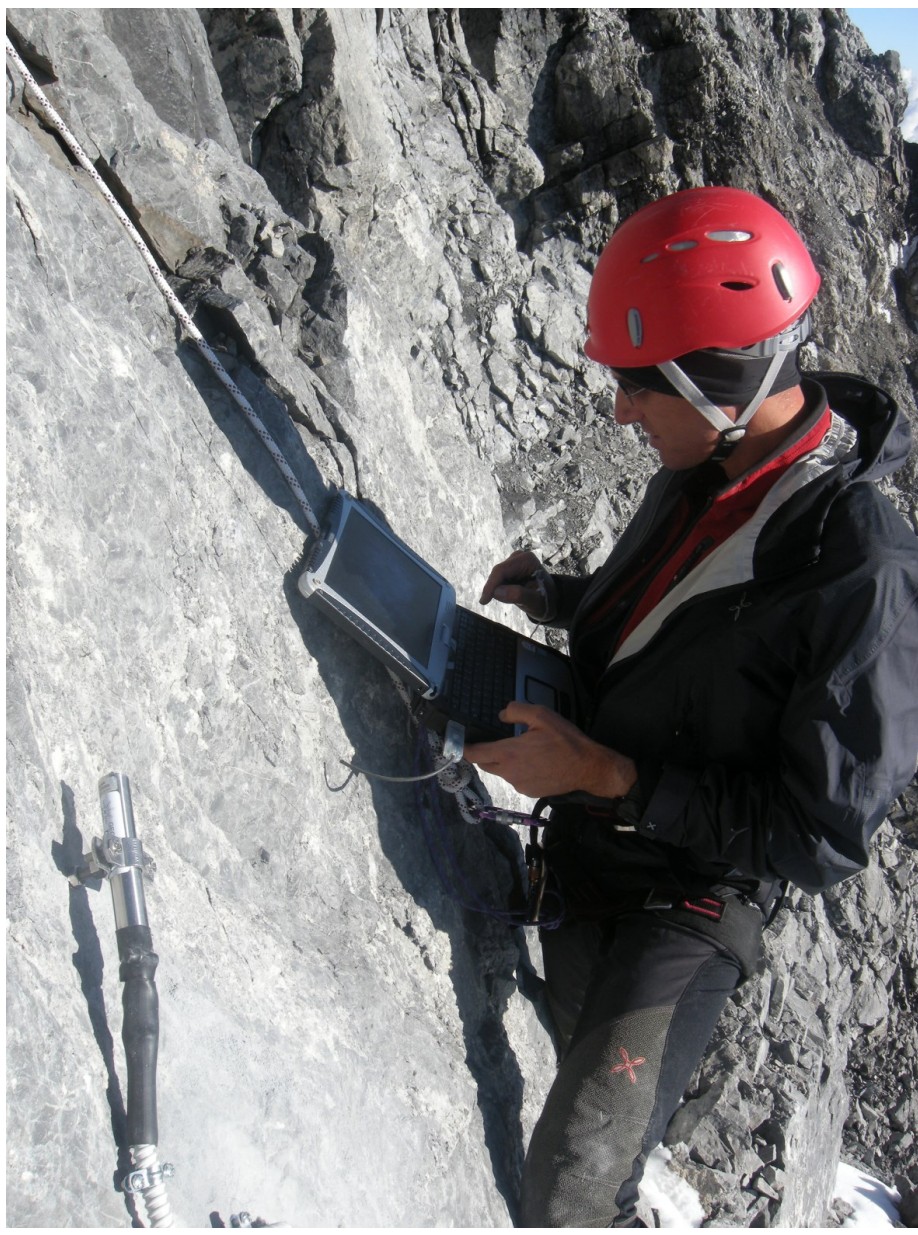

Figure I2. Launching of the CIMA ALTO data logger. A wireless USB dongle secures the wireless
connection to the laptop used for launching the logger and for downloading the temperature data. Photo
taken on 30 August 2011.






Figure I3. Location of the six sites equipped with data loggers for rock wall temperature measurement on Mt.

Ortles (BIV_LOMBARDI and HINTERGRAT from © Google Earth Pro 7.3 (2022)).



**Acknowledgements**
This work is a contribution to the Ortles Project, a program supported by two NSF awards no. 1060115 & no. 1461422
to The Ohio State University and by the Ripartizione Protezione Antincendi e Civile of the Autonomous province of
Bolzano in collaboration with the Ripartizione Opere idrauliche e Ripartizione Foreste of the Autonomous province of
Bolzano and the Stelvio National Park. This is Ortles Project publication 11 (www.ortles.org). The research was funded
by the Italian MIUR Project (PRIN 2010-11), "Response of morphoclimatic system dynamics to global changes and
related geomorphological hazards" (local and national coordinators G. Dalla Fontana and C. Baroni). The authors are
grateful to all the students, technicians and scientists who contributed to the field activities in the period from 2009 to
2016, the Alpine guides of the Alpinschule of Solda, the helicopter companies Airway, Air Service Center, Star Work
Sky and the Hotel Franzenshöhe for logistical support. The authors acknowledge the support of Vinicio Carraro (TeSAF
Department of the University of Padova) for the setup of the automatic weather station, and of Umberto Morra di Cella
and Paolo Pogliotti (ARPA Val d'Aosta) for the setup of the rock wall temperature dataloggers. The soil surface
temperature measured on Mt. Vioz were kindly provided by the Servizio Geologico of the Autonomous province of
Trento (Matteo Zumiani).

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
