# Peer review of "Modern air, englacial and permafrost temperatures at high altitude on Mt."

_Earth System Science Data, 2023_

## Referee Comment (RC1)

**General comments**

The authors set out to describe the modern air, englacial and permafrost temperatures on Mt. Ortles, in the Eastern European Alps. The manuscript by Carturan et al. is well-written paper. The data and instruments used are presented and described with sufficient detail. The data description is well explained and accurate, including complete and exhaustive images and tables.

The study site was well chosen with a dataset covered six years with only a few gaps for specific months. The data set is easily accessible through the link written in the manuscript (section 5. Data Availability).

Good narrative, flow and pause throughout the document. Overall, this manuscript is a relevant that deserves publication once the minor corrections are addressed. The minor suggestions are reported in the Specific Comments.

**Specific comments**

L 189: What is the range of these variations? Please provide some number.

L 281: The adjective "very" is subjective. Please, provide a degree range also in this case.

L 303: Range of the accuracy?

**Technical corrections**

Title: Please consider to remove the comma after "Ortles" and put it after the brackets.

L 36: …rock wall *temperatures*

L 37: …regarding instrument *types*

L 39: In the observed period,

L 45-47: Please reformulate the sentence. It is unclear.

L 84: In this paper, ……..englacial *temperatures*

L 87: make a unique sentence with the previous one "……Ortles Project *which* is an international…"

L 95-96: remove from the brackets 3905 m a.s.l. and create a unique sentence "….Italian Alps (Carturan et al., 2013) *and* it is the highest peak *(3905 m a.sl.)* of South Tyrol".

L 107: consider replace "with" with "so".

L 111: Insert the two ice bodies names

L 143: …..datasets *due to* the impossibility….

L 290-291: Please consider reformulating the sentence. Example: "…..but was repaired in late August 2012. It remained operational until August 2014 when it was removed due to another badly damage".

L 294: ……battery failure *while* the other….

L 336: are

L 343: delete "as such"

L 380: decreased

Figure 1: Panel d) Please consider leaving only "Hintergrat ridge". Insert the year of the DEM used in this Figure.

Figure 5: Consider splitting the y axis more.

Figure 7: Please consider inserting the legend also in the other panels (b, c, d, e and f) or put the present legend at the end of all panels.

Table 2: Consider to remove "for air temperature measurements". Also the sentence "Differences are calculated in the common working period for pairs of sensors, i.e. they refer to different period" should be along the main text and not here.

---

## Author Comment (AC1)

Dear Editor,

we would like to thank the two Reviewers for the careful reviews and for the suggestions that helped us improving considerably our manuscript.

We have addressed all the points highlighted by the reviewers and we modified the manuscript accordingly. In particular, our major changes in the revised version of the manuscript are:

- addition of more recent references in the Introduction Section
- addition of the new Figure 3
- remake of Figure 6 (former Figure 5)
- remake of Figure 8 (former Figure 7)
- addition of considerations regarding mean annual precipitation at the study site, in Section 2

In the following, we answer in detail to the specific comments made by the reviewers. The author responses are reported in italic right below the reviewers' comments. Line and page numbers are referred to the submitted paper.

**Reviewer 1**

General comments

The authors set out to describe the modern air, englacial and permafrost temperatures on Mt. Ortles, in the Eastern European Alps. The manuscript by Carturan et al. is well-written paper. The data and instruments used are presented and described with sufficient detail. The data description is well explained and accurate, including complete and exhaustive images and tables. The study site was well chosen with a dataset covered six years with only a few gaps for specific months. The data set is easily accessible through the link written in the manuscript (section 5. Data Availability). Good narrative, flow and pause throughout the document. Overall, this manuscript is a relevant that deserves publication once the minor corrections are addressed. The minor suggestions are reported in the Specific Comments.

Specific comments

L 189: What is the range of these variations? Please provide some number.

*Reply: we have added the mean and the 5$^{th}$ and 95$^{th}$ percentiles.*

L 281: The adjective "very" is subjective. Please, provide a degree range also in this case.

*Reply: we have added reference to Table E1 that reports slope angles.*

L 303: Range of the accuracy?

*Reply: we have added reference to Table 1, which reports the accuracy of each sensor.*

Technical corrections

Title: Please consider to remove the comma after "Ortles" and put it after the brackets.

*Reply: modified as suggested*

L 36: …rock wall temperatures

*Reply: modified as suggested*

L 37: …regarding instrument types

*Reply: modified as suggested*

L 39: In the observed period,

*Reply: modified as suggested*

L 45-47: Please reformulate the sentence. It is unclear.

*Reply: ok, rephrased*

L 84: In this paper, ……..englacial temperatures

*Reply: modified as suggested*

L 87: make a unique sentence with the previous one "……Ortles Project which is an international…"

*Reply: modified accordingly*

L 95-96: remove from the brackets 3905 m a.s.l. and create a unique sentence "….Italian Alps (Carturan et al., 2013) and it is the highest peak (3905 m a.sl.) of South Tyrol".

*Reply: ok, rephrased*

L 107: consider replace "with" with "so".

*Reply: ok, replaced and rephrased accordingly*

L 111: Insert the two ice bodies names

*Reply: ok glacier names added as suggested*

L 143: …..datasets due to the impossibility….

*Reply: modified accordingly*

L 290-291: Please consider reformulating the sentence. Example: "…..but was repaired in late August 2012. It remained operational until August 2014 when it was removed due to another badly damage".

*Reply: ok, rephrased as suggested*

L 294: ……battery failure while the other….

*Reply: modified accordingly*

L 336: are

*Reply: modified accordingly*

L 343: delete "as such"

*Reply: modified accordingly*

L 380: decreased

*Reply: modified accordingly*

Figure 1: Panel d) Please consider leaving only "Hintergrat ridge". Insert the year of the DEM used in this Figure.

*Reply: we would prefer to keep the figure as is because names are referred to Table 1. We have added the year of the DEM in the caption.*

Figure 5: Consider splitting the y axis more.

*Reply: we have modified this figure accordingly (2°C intervals instead of 5°C). Its name is now Figure 6.*

Figure 7: Please consider inserting the legend also in the other panels (b, c, d, e and f) or put the present legend at the end of all panels.

*Reply: we have put the legend at the end of all panels. Its name is now Figure 8.*

Table 2: Consider to remove "for air temperature measurements". Also the sentence "Differences are calculated in the common working period for pairs of sensors, i.e. they refer to different period" should be along the main text and not here.

*Reply: modified accordingly*

**Reviewer 2**

The manuscript by Carturan et al. presents a multi-year observational dataset of air temperature, englacial temperature and permafrost (soil and rockwall) temperatures at a high elevation site in the European Alps. The authors present a clear and well-written article describing the dataset, its quality control and some potential applications for the scientific community. The writing is clear and concise while providing sufficient detail to explain the datasets and their limitations. The dataset is highly valuable within the fields of glaciology and permafrost research as it provides a rare insight into processes occurring at high elevation, summit sites in the Alps. I have only a few minor comments/suggestions, but I would otherwise recommend this manuscript for publication in ESSD.

Specific comments:

L76-77: Can the authors also cite some more recent literature here? I think that it is useful to highlight that forcing uncertainty remains a strong challenge for high elevation, ungauged regions, especially where glaciers are located.

*Reply: ok, references added.*

L80: I think the authors can cite some more recent and highly relevant papers regarding the investigation of glacier cooling, particularly given the first-author's own work in the Ortles-Cevedale.

*Reply: ok, references added.*

L94: I find the site description section to be well-detailed. While the Appendix provides a clear overview of the instruments, I would personally like to see at least one of the photos of the Mt. Ortles station brought forward into the main text.

*Reply: ok, figure added (now Figure 3). Following figures have been renumbered consequently.*

L106-109: I understand that it is not the main goal of the manuscript to provide data on snowfall/precipitation at this site, but I think it would highlight the value of such high elevation stations if the authors could briefly describe how much snow might otherwise be estimated at this elevation using only low-elevation pluviometer stations or a single grid from reanalysis data. What about liquid precipitation at this site? Is there something to say about the relevance of the other data gathered at this station (e.g. the SR50) for constraining forcing uncertainty at high elevations?

*Reply: considerations on precipitation have been added in this section, as suggested.*

L174-191: The authors provide many air temperature observations and aptly describe their susceptibility to heating errors and issues related to the changing snow conditions. I feel that the authors could provide an additional figure to highlight the deviations between air temperature observations on an hourly basis and show the magnitudes of the heating effects for different conditions. Although a full analysis/parameterisation of this heating error is not necessary, I think such a figure would emphasise well a lot of potential data issues that the authors allude to in the text (e.g. L305-311). I do think that the QA flags provided in the data series are very helpful. Figure 3 is, however, not so informative for the impact on the temperature series, even if the resultant impact of the sensor height above snow on air temperature is mostly obvious.

*Reply: we think that this analysis is beyond the aim of the paper, and would require a joint analysis of all nivometeorological data collected on Mt. Ortles (for example wind speed, solar radiation, snow depth). In addition, this analysis is the topic of a separated manuscript, which is in preparation.*

Figure 7: Please correct the legend that reads the depths of rockwall temperatures in 'm', not 'cm' (cf. L287)

*Reply: ok, corrected.*